# Online Ad Allocation with Predictions

**Fabian Spaeh**
Department of Computer Science
Boston University
fspaeh@bu.edu

**Alina Ene**
Department of Computer Science
Boston University
aene@bu.edu

## Abstract

Display Ads and the generalized assignment problem are two well-studied online packing problems with important applications in ad allocation and other areas. In both problems, ad impressions arrive online and have to be allocated immediately to budget-constrained advertisers. Worst-case algorithms that achieve the ideal competitive ratio are known, but might act overly conservative given the predictable and usually tame nature of real-world input. Given this discrepancy, we develop an algorithm for both problems that incorporate predictions and can thus improve the performance beyond the worst-case. Our algorithm is based on the work of Feldman et al. (2009a) and similar in nature to Mahdian et al. (2007) who were the first to develop an algorithm with predictions for the related, but more structured Ad Words problem. We use a novel analysis to show that our algorithm is able to capitalize on a good prediction, while being robust against poor predictions. We experimentally evaluate our algorithm on synthetic and real-world data on a wide range of predictions. Our algorithm is consistently outperforming the worst-case algorithm without predictions.

## 1 Introduction

Advertising on the internet is a multi-billion dollar industry with ever growing revenue, especially as online retail gains more and more popularity. Typically, a user arrives on a website which fills an empty advertising spot (called an impression) by allocating it instantly to one of many advertisers. Advertisers value users differently based on search queries or demographic data and reveal their valuations in auctions or through contracts with the website. Formulations of increasing complexity have been studied to capture this problem, creating a hierarchy of difficulty (Mehta, 2013). The most basic is online bipartite matching, where each vertex on one side arrives online with all its adjacent edges and has to be matched immediately to one of the vertices on the other side, which were supplied offline. The problem and all generalizations admit a hard lower bound of $1 - \frac{1}{e}$ due to the uncertainty about future vertices. Motivated by online ad exchanges, where advertisers place bids on impressions, Mehta et al. (2007) introduced the Ad Words problem, which is a generalization of online bipartite matching where we charge each advertiser for the amount they bid. Going beyond the Ad Words setting, Feldman et al. (2009a) considered the more expressive problems Display Ads and the generalized assignment problem (GAP), and proposed algorithms for these settings with worst-case guarantees.

Classic algorithms that defend against the worst case of $1 - \frac{1}{e}$ are often overly conservative given that the real world does not behave like a contrived worst-case instance. Recently, researchers have thus been trying to leverage a prediction about some problem parameter to go beyond the worst-case (Mitzenmacher and Vassilvitskii, 2022). In the context of ad allocation, a prediction can be the keyword distribution of users on a certain day, or simply the advertiser allocation itself. Such a prediction is readily obtainable in practice, for example through learning on historic data. Two opposing properties are important: The algorithm has to be consistent, meaning that its performance

should improve with the prediction quality. Simultaneously, we want the algorithm to be robust against a poor prediction, i.e. not to decay completely but retain some form of worst-case guarantee. This is particularly important in the advertising business as much revenue is extracted from fat tails containing special events that are difficult or impossible to predict, but extremely valuable for the advertising business (e.g. advertising fan merchandise after a team's victory). To this end, Mahdian et al. (2007) developed an algorithm with predictions for the Ad Words problem and Medina and Vassilvitskii (2017) show how to use bid predictions to set reserve prices for ad auctions. Inspired by their work, we develop an algorithm with predictions for Display Ads and GAP.

**Our Contributions:** We design the first algorithms that incorporate predictions for the well-studied problems Display Ads and GAP. The two problems are general online packing problems, that capture a wide range of applications. Our algorithm follows a primal-dual approach, which yields a combinatorial algorithm that is very efficient and easy to implement. It is able to leverage predictions from any source, such as predictions constructed from historical data. Using a novel analysis, we show that the algorithm is robust against bad predictions and able to improve its performance with good predictions. In particular, we are able to bypass the strong lower bound on the worst-case competitive ratio for these problems. We experimentally verify the practical applicability of our algorithm under various kinds of predictions on synthetic and real-world data sets. Here, we observe that our algorithm is able to outperform the baseline worst-case algorithm due to Feldman et al. (2009a) that does not use predictions, by leveraging predictions that are obtained from historic data, as well as predictions that are corrupted versions of the optimum allocation.

## 1.1 Preliminaries

**Problem Definition:** In this work, we study the Display Ads problem and its generalization, the generalized assignment problem (GAP) (Feldman et al., 2009a). In Display Ads, there are advertisers $a \in \{1, \ldots, k\}$ that are known ahead of time, each of which is willing to pay for at most $B_a$ ad impressions. A sequence of ad impressions arrive online, one at a time, possibly in adversarial order. When impression $t$ arrives, the values $w_{at} \geq 0$ for each advertiser $a$ become known. These values might be a prediction of click-through probability or any valuation from the advertiser, but we treat them as abstractly given to the algorithm. We have to allocate $t$ immediately to an advertiser, or decide not to allocate it at all. The goal is to maximize the total value $\sum_{a,t} x_{at} w_{at}$ subject to $\sum_t x_{at} \leq B_a$ for all $a$, where $x_{at} = 1$ if $t$ is allocated to $a$ and $x_{at} = 0$, otherwise. GAP is a generalization of Display Ads where the size that each impression takes up in the budget constraint of an advertisers is non-uniform. That is, each impression $t$ has a size $u_{at} \geq 0$ for each advertisers $a$, and advertiser $a$ is only willing to pay for a set of impressions whose total size is at most $B_a$. More precisely, we require that $\sum_{a,t} x_{at} u_{at} \leq B_a$.

**Free Disposal Model:** In general, it is not possible to achieve any competitive ratio for the online problems described above. Motivated by online advertising, Feldman et al. (2009a) introduced the free disposal model which makes the problem tractable: when a new impression arrives, the algorithm allocates it to an advertiser $a$; if $a$ is out of budget, we can decide to dispose of an impression previously allocated to $a$. The motivation for this model is that advertisers are happy to receive more ads, as long as they are only charged for the $B_a$ most valuable impressions. We refer the reader to the paper of Feldman et al. (2009a) for additional motivation of this model. In this work, we consider both Display Ads and GAP in the free disposal model.

**Related Problems:** Display Ads and GAP are significant generalizations of well-studied problems such as online bipartite matching and Ad Words. In online bipartite matching, all values are $1$. In Ad Words, values and sizes are identical. The latter setting allows for more specialized algorithms that exploit the special properties of this problem, as we discuss in more detail later.

**Algorithms with Predictions:** The algorithms we study follow under the umbrella of algorithms that leverage predictions to obtain improved performance. These were studied in an extensive line of work, see e.g. the survey of Mitzenmacher and Vassilvitskii (2022). Following this established research, we use two important measures for the performance of the algorithm: The *robustness* ALG/OPT indicates how well the algorithm's objective value ALG performs against the optimum solution OPT; the *consistency* ALG/PRD measures how close the algorithm gets to the prediction's objective value PRD. Most algorithms with predictions, including the one presented in this work, allow to control the trade-off between robustness and consistency with a parameter $\alpha$.

## 1.2 Related Work

**Online Ad-Allocation with Predictions:** To the best to our knowledge, we are the first to study Display Ads and GAP with predictions. Related problems were considered in the work by Mahdian et al. (2007) and Medina and Vassilvitskii (2017) for Ad Words, and by Lattanzi et al. (2020) for online capacitated bipartite matching. Medina and Vassilvitskii (2017) use bid predictions to set reserve prices for ad auctions. Lavastida et al. (2021) incorporate predictions of the dual variables into the proportional-weights algorithm (Agrawal et al., 2018; Karp et al., 1990; Lattanzi et al., 2020) for online capacitated bipartite matching. Chen and Indyk (2021) analyze an algorithm that uses degree predictions by matching arriving vertices to vertices with minimum predicted degree. Jin and Ma (2022) provide an algorithm for batched allocation in bipartite matching under predictions, based on the model of Feng et al. (2021). As noted above, Ad Words and bipartite matching have additional structure, which is exploited in these prior works. In particular, the algorithms proposed in these works are not applicable to the more general problems Display Ads and GAP. Our algorithm builds on the approach of Mahdian et al. (2007) for the Ad Words problem, but substantial new ideas are needed in the algorithm design and analysis, as discussed in more detail in Section 2.

There has been further extensive work in the design of worst-case algorithms and under random input models without predictions, which we now summarize.

**Worst-Case Algorithms:** The design of worst-case algorithms has been the focus of a long line of work which can, for instance, be found in the survey of Mehta (2013). A large focus has been on Ad Words. Several combinatorial algorithms have been proposed, based on the work of Karp et al. (1990). The combinatorial approach is tailored to the structure of to these special cases. The primal-dual approach is a more general approach that can handle more complex problems such as Display Ads and GAP (Buchbinder et al., 2007; Feldman et al., 2009a). In this work, we build on the primal-dual algorithm of Feldman et al. (2009a) and show how to incorporate predictions into their framework. The worst-case guarantee for online bipartite-matching, and therefore for all generalizations, is $1 - \frac{1}{e}$ (Karp et al., 1990).

**Stochastic Algorithms:** The lower bound of $1 - \frac{1}{e}$ can be circumvented under distributional assumptions. This has been extensively studied for online bipartite matching (Karande et al., 2011; Feldman et al., 2009b; Jin and Williamson, 2022). Further work has been done for the Ad Words problem (Devanur and Hayes, 2009; Devanur et al., 2012) with generalizations due to Feldman et al. (2010) for a more general stochastic packing problem.

## 2 Our Algorithm

In order to illustrate the algorithmic ideas and analysis, we consider the simpler setting of Display Ads in this section. Our algorithm for GAP is a generalization of this algorithm and we include it in Appendix B. For simplicity, we assume that our prediction is a solution to the problem, which is as in prior work (Mahdian et al., 2007). However, due to our general analysis framework, we also consider our algorithm a starting point towards incorporating weaker predictors, such as partial solutions or predictions of the supply, which we leave for future work.

**Prediction:** We assume that we are given access to a prediction, which is a fixed solution to the problem, given as an allocation of impressions to advertisers. With each impression $t$, we also receive the advertiser $\text{PRD}(t)$ to which the prediction allocates $t$. In particular, this means that the prediction does not have to be created up front, but can be adjusted on the fly based on the observed impressions.

Given a solution to the problem, we could also consider the following random-mixture algorithm: For some parameter $q \in [0, 1]$, run the worst-case algorithm; with probability $1 - q$, follow the prediction exactly. This algorithm achieves a robustness of $q \cdot (1 - \frac{1}{e})$ and consistency of $q \cdot (1 - \frac{1}{e}) + 1 - q$. However, this is only in expectation and against a weak adversary that is oblivious to the algorithm's random choices. In contrast, Algorithm 1 is designed to obtain its guarantees deterministically against the strongest possible adversary that can adapt to the algorithm's choices, which is identical to the setup in Mahdian et al. (2007). We observe that Algorithm 1 clearly outperforms this random-mixture algorithm in our experiments (cf. Section 4) which shows that our stronger setting is indeed valuable in practice. Furthermore, the random-mixture algorithm cannot be adapted to different predictors that are not solutions, such as the ones mentioned above.

**Algorithm Overview:** Our Algorithm, shown in Algorithm 1, incorporates predictions in the

| Display Ads Primal | Display Ads Dual |
|---|---|

$$\max \sum_{a,t} w_{at} x_{at} \qquad\qquad \min \sum_{a} B_a \beta_a + \sum_t z_t$$

$$\forall a:\ \sum_t x_{at} \leq B_a \qquad\qquad \forall a,t:\ z_t \geq w_{at} - \beta_a$$

$$\forall t:\ \sum_a x_{at} \leq 1 \qquad\qquad \forall a:\ \beta_a \geq 0$$

$$\qquad\qquad\qquad \forall t:\ z_t \geq 0$$

$$\forall a,t:\ x_{at} \geq 0$$

Figure 1: Primal and dual of the Display Ads LP

---

**Algorithm 1** Exponential Averaging with Predictions
___
1: **Input:** Robustness-consistency trade-off parameter $\alpha \in [1, \infty)$, advertiser budgets $B_a \in \mathbb{N}$
2: Define the constants $B := \min_a B_a$, $e_B := \left(1 + \frac{1}{B}\right)^B$, and $\alpha_B := B\left(e_B^{\alpha/B} - 1\right)$
3: For each advertiser $a$, initialize $\beta_a \leftarrow 0$ and allocate $B_a$ zero-value impressions
4: **for all** arriving impressions $t$ **do**
5: $\quad a_{(\mathrm{PRD})} \leftarrow \mathrm{PRD}(t)$
6: $\quad a_{(\mathrm{EXP})} \leftarrow \arg\max_a \{w_{at} - \beta_a\}$
7: $\quad$**if** $\alpha_B(w_{a_{(\mathrm{PRD})}t} - \beta_{a_{(\mathrm{PRD})}}) \geq w_{a_{(\mathrm{EXP})}t} - \beta_{a_{(\mathrm{EXP})}}$ **then**
8: $\quad\quad a \leftarrow a_{(\mathrm{PRD})}$
9: $\quad$**else**
10: $\quad\quad a \leftarrow a_{(\mathrm{EXP})}$
11: $\quad$**end if**
12: $\quad$Allocate $t$ to $a$ and remove the least valuable impression currently assigned to $a$
13: $\quad$Let $w_1 \leq w_2 \leq \cdots \leq w_{B_a}$ be the values of impressions currently assigned to $a$ in non-decreasing order
14: $\quad$Update $\beta_a \leftarrow \dfrac{e_{B_a}^{\alpha/B_a} - 1}{e_{B_a}^{\alpha} - 1} \sum_{i=1}^{B_a} w_i e_{B_a}^{\alpha(B_a - i)/B_a}$
15: **end for**
___

primal-dual algorithm of Feldman et al. (2009a). The algorithm is based on the primal and dual LP formulations in Figure 1. The algorithm constructs both a primal integral solution which is an allocation of impressions to advertisers as well as a dual solution, given explicitly by the dual variables $\beta_a$. Analogously to other algorithms with predictions, the algorithm takes as parameter the value $\alpha$; a larger value of $\alpha$ means that we trust the prediction more. Similarly to the worst-case algorithm of Feldman et al. (2009a), the dual variables $\beta_a$ play an important role in the allocation of impressions. When an impression $t$ arrives, we evaluate the discounted gain $w_{at} - \beta_a$ for each advertiser. The worst-case algorithm allocates the impression to the advertiser $a_{(\mathrm{EXP})}$ maximizing the discounted gain and only allocates if the discounted gain is positive, i.e. the value exceeds the threshold $\beta_a$. Our algorithm modifies this base algorithm to incorporate predictions as shown in Line 7 and it follows the prediction $a_{(\mathrm{PRD})}$ if its discounted gain is a sufficiently high fraction of the discounted gain of $a_{(\mathrm{EXP})}$. We refer the reader to the discussion below for more intuition on the choice of update. After selecting the advertiser to which to allocate the impression $t$, we remove the least valuable impression currently assigned to $a$ to make space for $t$, and then allocate $t$ to $a$. Another crucial part of the algorithm is the update rule for $\beta_a$ in Line 14, which is updated in a novel way based on the parameter $\alpha$. More precisely, we set $\beta_a$ as an exponential averaging of the values of impressions currently allocated to $a$. Compared to the worst-case algorithm, we assign higher weight to impressions with less value which is essential for leveraging predictions.

To simplify the algorithm description and analysis, we initially allocate $B_a$ impressions of zero value to each advertiser $a$. Furthermore, we assume that there exists a "dummy" advertiser with large budget that only receives zero value impressions. Instead of not allocating an impression explicitly (either in the algorithm or the prediction), we allocate to the dummy advertiser, instead.

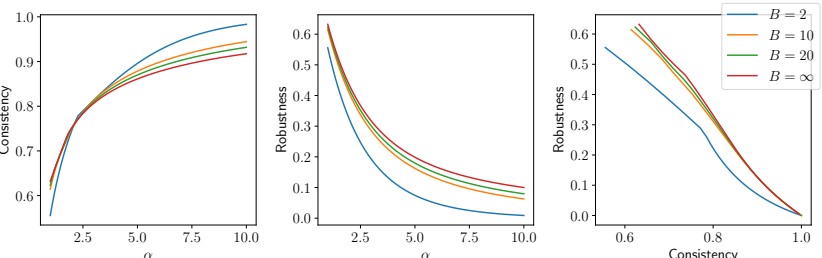

Figure 2: We illustrate the consistency-robustness trade-off of Algorithm 1 for various values of $\alpha$ and budgets $B$.

**Intuition for our Algorithm:** As noted above, we make two crucial modifications to the worst-case algorithm to incorporate predictions: The advertiser selection (Line 7) and the update of $\beta_a$ (Line 14). We now provide intuition for both choices.

First, let us illustrate the difficulties in incorporating predictions in the advertiser selection. Based on the worst-case algorithm, a natural way to incorporate predictions is to allocate to the prediction if the distorted gain $w_{a_{(\text{PRD})}t} - \frac{1}{\alpha} \cdot \beta_{a_{(\text{PRD})}}$ exceeds the maximum discounted gain. However, this approach does not work as shown by the following example: Consider a scenario where the prediction suggests a constant advertiser $a_{(\text{PRD})}$. Impressions are split into two phases: Phase 1 contains $B_{a_{(\text{PRD})}}$ impressions $t$ where only $a_{(\text{PRD})}$ can derive a value of $w_{a_{(\text{PRD})}t} = 1$ and $w_{at} = 0$ for $a \neq a_{(\text{PRD})}$. The algorithm allocates all these impressions to $a_{(\text{PRD})}$ and at the end of phase 1 has $\beta_{a_{(\text{PRD})}} = 1$. In phase 2, a large amount of impressions with $w_{a_{(\text{PRD})}t} = 1$ and $w_{at} = \frac{\alpha-1}{2\alpha}$ for $a \neq a_{(\text{PRD})}$ arrive. Since the distorted gain $w_{a_{(\text{PRD})}t} - \frac{1}{\alpha}\beta_{a_{(\text{PRD})}} = \frac{\alpha-1}{\alpha}$ exceeds the discounted gain $w_{at} - \beta_a = \frac{\alpha-1}{2\alpha}$ for $a \neq a_{(\text{PRD})}$, the algorithm allocates to $a_{(\text{PRD})}$ which yields 0 gain. However, we forfeit an unbounded amount of potential value derived from allocating to advertisers $a \neq a_{(\text{PRD})}$. An important takeaway of this example is the crucial observation that the algorithm should never allocate to the predicted advertiser if its discounted gain is 0. The selection rule in our algorithm is designed to meet this important consideration.

Second, we need to change the update rule for $\beta_a$. We update $\beta_a$ using a carefully selected exponential average of the values of impressions currently assigned to $a$, that incorporates our confidence in the prediction parameterized by $\alpha$. In contrast to the worst-case algorithm, we weigh less valuable impressions more. This lowers the threshold for the addition of new impressions, which allows us to exploit more potential gain from the predicted advertiser.

**Theorem 1.** *Let $B := \min_a B_a$ and $e_B := \left(1 + \frac{1}{B}\right)^B$. Let OPT and PRD be the values of the optimal and predicted solutions, respectively. For any $\alpha \geq 1$, Algorithm 1 obtains a value of at least*

$$\text{ALG} \geq \max\left\{R(\alpha) \cdot \text{OPT}, C(\alpha) \cdot \text{PRD}\right\}$$

*where the robustness is*

$$R(\alpha) := \frac{e_B^\alpha - 1}{B e_B^\alpha \left(e_B^{\alpha/B} - 1\right)} \to \frac{e^\alpha - 1}{\alpha e^\alpha} \quad (B \to \infty)$$

*and the consistency is*

$$C(\alpha) := \left(1 + \frac{1}{e_B^\alpha - 1} \max\left\{\frac{1}{\alpha B}\left(e_B^\alpha - \frac{e_B^\alpha - 1}{\alpha B}\right), \ln\left(e_B^\alpha\right)\right\}\right)^{-1}$$

$$\to \left(1 + \frac{1}{e^\alpha - 1} \max\left\{\frac{1}{\alpha}\left(e^\alpha - \frac{e^\alpha - 1}{\alpha}\right), \alpha\right\}\right)^{-1} \quad (B \to \infty).$$

Figure 2 shows the above trade-off between consistency and robustness. We can observe that the guarantee rapidly improves as the minimum advertiser budget $B$ increases, which is very beneficial both in theory and in practice. The trade-off is comparable to the one obtained by Mahdian et al. (2007) for the more structured Ad Words problem.

**Comparison to Prior Work:** The work most closely related to ours is the work of Mahdian et al. (2007) which incorporates predictions into the algorithm of Mehta et al. (2007). These works are for the related but different Ad Words problem, and do not apply to Display Ads and GAP. In the Ad Words problem, the value of an impression is equal to the price that the advertiser pays for it, i.e., the size of the impression in the budget constraint. In contrast, in Display Ads and GAP, the values and the sizes are independent of each other (e.g., in Display Ads an impression takes up 1 unit of space in the advertiser's budget but it can accrue an arbitrary value). Thus there is no longer any relationship between the total value/profit of the impressions and the amount of the advertiser's budget that has been exhausted. The Ad Words algorithms of Mehta et al. (2007); Mahdian et al. (2007) crucially rely on this relationship both in the algorithm and in the analysis. Due to the special structure of the problem, these algorithms do not dispose of impressions and consider only the fraction of the advertiser's budget that has been filled up in order to decide the allocation and to incorporate the prediction. Moreover, the algorithm and the analysis do not need to account for the loss incurred by disposing impressions. These crucial differences require a new algorithmic approach and analysis, which was given by Feldman et al. (2009a) using a primal-dual approach. Since we build on their framework as opposed to the work of Mehta et al. (2007), we also need a new approach for incorporating predictions, as described above in the intuition for our algorithm. Further, the primal-dual framework only helps in establishing the robustness but not the consistency of our algorithm, and we develop a novel combinatorial analysis for proving the consistency.

## 3 Analysis

In the following, we outline the analysis of Algorithm 1 to prove Theorem 1. Specifically, we show separately that $\mathrm{ALG}/\mathrm{OPT} \leq R(\alpha)$ (robustness) and $\mathrm{ALG}/\mathrm{PRD} \leq C(\alpha)$ (consistency).

**Notation:** We denote with superscript $(t)$ the value of variables after allocating impression $t$. E.g. $a^{(t)}$ is the algorithm's choice of advertiser for impression $t$ and $\beta_a^{(t)}$ is the dual variable after allocating $t$. Let

$$\mathbf{X}_a \coloneqq \left\{t : a^{(t)} = a\right\} \quad \text{and} \quad \mathbf{P}_a \coloneqq \left\{t : a_{(\mathrm{PRD})}^{(t)} = a\right\}$$

be the impressions that were assigned to $a$ and potentially disposed of, and the impressions that the prediction recommended for assignment to $a$, respectively. We set $I_a \coloneqq |\mathbf{X}_a|$ and $\ell_a \coloneqq |\mathbf{P}_a \cap \mathbf{X}_a|$ as the size of the overlap. Let also $T$ be the last impression and $\mathbf{S}_a$ is the final allocation, i.e., the $B_a$ impressions allocated to $a$ at the end of the algorithm. Finally, let ALG and PRD be the total value of the solution created by the algorithm and the prediction, respectively.

**Robustness:** Our proof for robustness closely follows the analysis in Feldman et al. (2009a) by using the primal-dual formulation of the problem, with some additional care that is needed not to violate dual feasibility whenever we follow the prediction. We defer the full proof to Appendix A.1.

**Consistency:** We now show the consistency, i.e. that PRD is bounded by a multiple of ALG. The complete analysis can be found in Appendix A.2, while we only give a high-level overview here.

A common approach in the analysis of online primal-dual algorithms is to employ a local analysis where, in each iteration, we relate the change in the value of the primal solution to the change in the dual solution (Buchbinder and Naor, 2009). However, it is not clear how to employ such a strategy in our setting due to the complications arising from our algorithm following a mixture of the worst-case and predicted solutions. We overcome this challenge using a novel global analysis that relates the final primal value to the prediction's value.

We now provide a high level overview of our global analysis. We start by noting that the objective value of our algorithm and the prediction is the sum of the impression values allocated to each advertiser, i.e.

$$\mathrm{ALG} = \sum_a \sum_{t \in \mathbf{S}_a} w_{at} \quad \text{and} \quad \mathrm{PRD} = \sum_a \sum_{t \in \mathbf{P}_a} w_{at}$$

However, note that PRD contains values of impression that do not appear in ALG since we ignore the prediction in some iterations, or already disposed of the impression. It is further unclear which advertiser to "charge" for an impression that does not agree with the prediction.

Consider an impression for which we followed the worst-case choice $a_{(\mathrm{EXP})}$ that maximizes the discounted gain instead of the prediction. Due to our selection rule, the reason for this departure is

due to the discounted gains satisfying the following key inequality:

$$w_{a(\text{PRD})} \leq \frac{1}{\alpha_B} \left( w_{a(\text{EXP})} - \beta_{a(\text{EXP})} \right) + \beta_{a(\text{PRD})}. \tag{1}$$

By using this important relationship, we upper bound the value PRD of the prediction using a linear combination of the values of impressions allocated by the algorithm (but possibly disposed of) and the thresholds. By grouping the impression values and dual variables by advertiser in the resulting upper bound, we are able to correctly charge each impression for which we deviated from the prediction to a suitable advertiser, thus overcoming one of the challenges mentioned above. To summarize, using (1) we obtain a bound $\text{PRD} \leq \sum_a \text{PRD}_a$ where each $\text{PRD}_a$ is a linear combination of impression values and dual variables for advertiser $a$, and we want to compare this quantity to $\text{ALG}_a := \sum_{t \in S_a} w_{at}$.

Let us now consider a fixed advertiser $a$, and relate $\text{PRD}_a$ to $\text{ALG}_a$. At this point, a key difficulty is that the amount $\text{PRD}_a$ that we charged to advertiser $a$ involves the threshold $\beta_a$. By definition, the threshold is a convex combination of the values of the impressions in the algorithm's allocation. This gives us that $\text{PRD}_a$ is a linear combination of only the weights, but this cannot be readily compared to $\text{ALG}_a$ due to the complicated structure of the coefficients in the former. To bridge this gap, we show a useful structural property (Lemma 4) that gives us the following upper bound on $\text{PRD}_a$: If we define $t_i$ as the $i$-th impression allocated to $a$, we have

$$\text{PRD}_a \leq \sum_{i=I_a-B_a+1}^{I_a-\ell_a} \phi_i w_{at_i} + \sum_{i=I_a-\ell_a+1}^{I_a} \psi_i w_{at_i} + w_{at_{I_a-B_a}} \Omega_a \tag{2}$$

for appropriate coefficients $\phi_i$, $\psi_i$, and $\Omega_a$. The RHS of this inequality accounts for the value as follows: the first sum is for the impressions that agree with the prediction; the second sum is for the impressions that disagree with the prediction; the final term accounts for the values of all impressions that were disposed. As this is (almost) a linear combination over impression values that all appear in ALG (except $w_{at_{I_a-B_a}}$), we could bound the ratio $\text{PRD}_a/\text{ALG}_a$ by the maximum coefficient in (2). However, this does not lead to a constant competitive ratio. We therefore need a delicate analysis (Lemma 10) to balance the coefficients as uniformly as possible among all values, where we use properties of the coefficients $\phi_i$, $\psi_i$, and $\Omega_a$ and the structural property we derived in Lemma 4.

In Appendix B, we discuss how to generalize Algorithm 1 and its analysis to GAP, where impressions can take up arbitrary size $u_{at}$ in the budget constraints. This generalization uses a gain $w_{at} - u_{at}\beta_a$ that considers the size of each impression, and the comparison of the worst-case and predicted choices are updated correspondingly. We also change the update of $\beta_a$ to consider the impression density $w_{at}/u_{at}$. We also note that our technique can be used to achieve a slight improvement in the guarantees, but we omit this in the interest of simplicity and conciseness.

## 4 Experimental Evaluation

We now evaluate the practical applicability of Algorithm 1. We use multiple baseline algorithms on different kinds of predictions, which we describe below. We showcase results on real-world and synthetic data, with further experimental results in Appendix C.

**Algorithms:** We compare Algorithm 1 to the random-mixture algorithm described in Section 2 (Random Mixture) and the worst-case algorithm without predictions due to Feldman et al. (2009a) (Worst-Case Baseline). Additionally, we consider two modifications of the worst-case algorithm: One allocates to the impression of maximum value (Greedy Baseline) and the other to the impression with maximum gain after disposal (Discounted Greedy Baseline).

**Predictors:** We consider variations of the following predictors. Recall that each predictor is a fixed allocation of impressions to advertisers that is revealed online.

(1) *Optimum Solution (OPT):* The optimum solution is obtained by solving the problem optimally offline using an LP solver. To evaluate our algorithm's robustness, we also consider a version of the optimum solution where a random $p$-fraction of the allocations has been corrupted. Under a *random corruption*, we corrupt by reallocating to randomly chosen advertisers. For a *biased corruption*, we sample a random permutation offline and corrupt by reallocating according to this permutation, generating a more adversarial corruption.

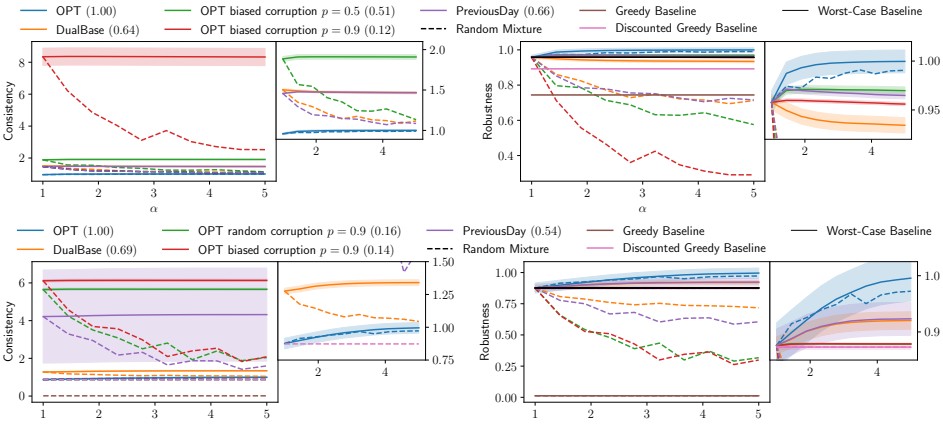

Figure 3: Experimental results on iPinYou (top) and Yahoo datasets (bottom) using different predictions for varying $\alpha$. The solid lines show our algorithm and the dashed lines the random-mixture algorithm. We run the algorithms 5 times and report average for both algorithms and the standard deviation only for our algorithm, to avoid clutter. For the robustness, the black line shows the performance of the worst-case algorithm without predictions due to Feldman et al. (2009a). For each predictor, we also include in parentheses the average competitive ratio over 5 runs (e.g. PreviousDay (0.66) indicates that the average competitive ratio for the solution of the Previous Day prediction was 0.66). We run the random-mixture algorithm for each prediction and $q := 1/\alpha$.

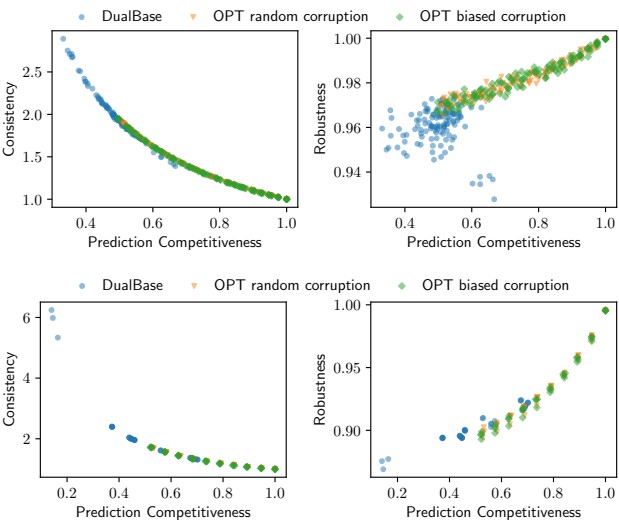

Figure 4: Performance of our algorithm on the iPinYou (top) and Yahoo (bottom) datasets for $\alpha = 5$ with predictions of varying quality obtained as follows: We vary the sample fraction $\epsilon \in [0, 1]$ for the dual base algorithm and $p \in [0, 1]$ for random and biased corruptions. The prediction competitiveness is defined as PRD/OPT.

(2) *Dual Base:* We generate a solution using the algorithm of Devanur and Hayes (2009). Here, we sample the initial $\epsilon$-fraction of all impressions and optimally solve a scaled version of the dual LP to obtain the dual variables $\{\beta_a\}_a$. We get a primal allocation for all future impressions $t$ by allocating to the advertiser $a$ that maximizes the discounted gain $w_{at} - \beta_a$, but do not update $\beta_a$.

(3) *Previous Day:* We look at all impressions from the previous day and optimally solve the dual LP offline to obtain dual variables $\{\beta_a\}_a$. To get a prediction for today's impressions, we use the same algorithm as above and allocate to the advertiser maximizing the discounted gain.

**Real-World Instances:** We generate two instances for Display Ads based on the real-word datasets iPinYou (Zhang et al., 2014) and Yahoo (Yahoo, 2011).

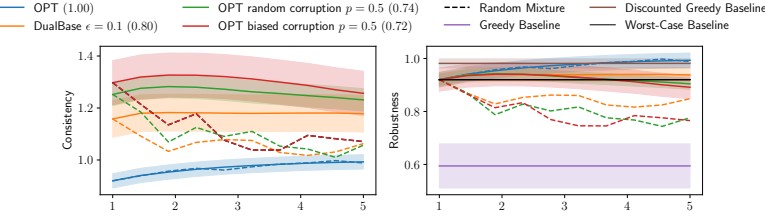

Figure 5: Experimental results for varying values of $\alpha$ on synthetic data with 12 advertisers and 2000 impressions of 10 types, where we report the same quantities as in Figure 3. We use different predictors with $\sigma = 1.5$. We conducted experiments with significantly more or less impressions, where we obtain similar results, and we therefore omit these.

(1) *iPinYou:* The iPinYou dataset contains real-time bidding data from the iPinYou online advertisement exchange. This dataset contains 40372 impressions over 7 days and 301 advertisers. Each advertiser places multiple bids for an impression. We use this bid data to construct the dataset. Specifically, we set the maximum of those bids as the advertiser's valuation. We assume a constant budget for each advertiser of 10 impressions as it makes for an interesting instance.

(2) *Yahoo:* We replicate the experimental setup of Lavastida et al. (2021) who generated an instance of online capacitated bipartite matching based on a Yahoo dataset (Yahoo, 2011). Capacitated online bipartite matching is a special case of Display Ads where all impression values are 1. Based on this dataset, we create an instance of capacitated online bipartite matching with around 2 million impressions and 16268 advertisers for 123 days. We defer the details to Section C.1.

**Synthetic Instances:** We obtain random synthetic data for a fixed set of $T$ impressions and $k$ advertisers as follows. We first generate a set of impression types, whereas each type is meant to model a group of homogeneous users (e.g. similar demographic or using similar keywords) and advertisers value users from the same group identically. We sample an advertiser's valuation for each impression type from an exponential distribution. To represent a full day of impressions, we assume that display times of impressions from a certain type are distributed according to a Gaussian with some uniformly random mean in $[0, 1]$ and a fixed standard deviation $\sigma$. We then sample the same number of impressions from each type along with display times, and order them in increasing display time. Finally, we equip each advertiser with some fixed budget that makes for a difficult instance.

**Results:** Figure 3 and Figure 5 show results for real-world and synthetic instances, respectively. For each predictor, we show the consistency (left) and robustness (right) for varying $\alpha$. Figure 4 shows results for $\alpha = 5$ with predictions of different quality, as described in the figure caption.

**Discussion:** We make several observations. On real-world instances, there is only a single prediction for which the performance of our algorithm drops below the worst-case algorithm, even for heavily corrupted predictions. E.g., on the iPinYou dataset, our algorithm is still able to leverage a prediction with a corruption rate as high as $p = 50\%$, and improve upon the worst-case algorithm (see the green and black lines in the top right plot of Figure 3). Moreover, for higher performing predictors, the improvement over the worst-case algorithm is significantly higher in both datasets. See for example, the Previous Day predictions on the iPinYou dataset (the purple and black lines in the top right plot of Figure 3) or Dual Base predictions on the Yahoo dataset (the orange and black lines in the bottom right plot of Figure 3). The increase in robustness is surprising as it does not follow the behavior of our theoretical bounds (cf. Figure 2) which hold against an adversarial prediction. This suggests that on real-world data, our algorithm is able to exploit "good suggestions" from the prediction while ignoring "bad suggestions". Second, as we can see in Figure 3, the consistency of our algorithm for predictors except the optimum is always above 1, and is significantly high for artificially corrupted predictions. The robustness of our algorithm remains high in almost all cases, even for the most heavily corrupted predictions (cf. the right side of Figure 4). On synthetic instances, we observe that our algorithm is robust against both random and biased corruption, as the robustness does not drop to the prediction's low competitiveness of $\approx 0.7$. Furthermore, our algorithm performs well in combination with the dual base prediction for $\epsilon = 0.1$ (the orange line in Figure 5), even though the first 200 impressions are not representative of all impressions. On all instances, we clearly outperform the random-mixture algorithm which merely interpolates between the objective values of the worst-case algorithm and the prediction.

## Conclusion

We introduce a novel algorithm with predictions for Display Ads and GAP with free disposal. Our algorithm is based on the primal-dual approach and can be efficiently implemented in practice. We show its robustness using the primal-dual method similar to Feldman et al. (2009a) and use a novel combinatorial proof to show its consistency. Finally, our experiments show the applicability of our algorithm, which is able to improve beyond the worst-case performance using readily available predictions. An interesting direction left open by our work is to understand the optimal trade-off between robustness and consistency for Display Ads and GAP. This challenging direction has been explored recently in the works (Jin and Ma, 2022; Wei and Zhang, 2020) for a related problem. **Limitations:** Our algorithm requires a strong prediction that is a solution to the problem. We leave weaker predictions, such as partial solutions or predictions of the supply, for future work.

## Acknowledgments

FS and AE were supported in part by NSF CAREER grant CCF-1750333, NSF grant III-1908510, and an Alfred P. Sloan Research Fellowship.

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

## A  Omitted Proofs

We will need the following helper Lemma in the proofs of consistency and robustness.

**Lemma 2.** *Recall that* $e_B := \left(1 + \frac{1}{B}\right)^B$ *and* $\alpha_B := B\left(e_B^{\alpha/B} - 1\right)$. *We have*

1. $e_B \leq e_{B_a}$ *and*

2. $\alpha_B \geq \alpha_{B_a}$.

*Proof.* It is well known that $e_B$ converges to $e$ from below for $B \to \infty$. Furthermore, we can show that $\alpha_B$ is decreasing in $B$ for all $\alpha \geq 1$ by taking the derivative

$$\frac{\partial}{\partial B}\alpha_B = \left(1 + \frac{1}{B}\right)^\alpha - 1 - \frac{\alpha}{B}\left(1 + \frac{1}{B}\right)^{\alpha-1} = \left(1 + \frac{1}{B}\right)^{\alpha-1}\left(1 + \frac{1}{B}(1-\alpha)\right) - 1$$

$$\leq \left(1 + \frac{1}{B}\right)^{\alpha-1}\left(1 + \frac{1}{B}\right)^{1-\alpha} - 1 = 0$$

where the bound follows from Bernoulli's inequality, which states that $1 + rx \leq (1+x)^r$ for $x \geq -1$ and $r \in \mathbb{R} \setminus (0,1)$. $\square$

### A.1  Proof of Theorem 1 (Robustness)

We write $P$ and $D$ to denote the objective value of the primal and dual solutions, i.e. $P = \sum_a \sum_{t \in \mathbf{S}_a} w_{at}$ and $D = \sum_a B_a \beta_a + \sum_t z_t$ where $z_t$ is specified in the following proof to ensure feasibility. We can show that after the allocation of each impression $t$,

$$\Delta P \geq \frac{e_B^\alpha - 1}{Be_B^\alpha\left(e_B^{\alpha/B} - 1\right)}\Delta D$$

where $\Delta P$ and $\Delta D$ are the increase in the primal and dual solution values, respectively. Since we create feasible primal and dual solutions, this is sufficient to bound the robustness due to weak duality. There is one main difference to Feldman et al. (2009a): In their algorithm, setting the dual variable $z_t$ to $w_{a_{(\mathrm{EXP})}t} - \beta_a$ ensures dual feasibility as $a_{(\mathrm{EXP})}$ is the advertiser with maximum discounted gain. However, in order not to violate dual feasibility when following the prediction, we need to increase the dual variables $z_t$ by a factor of $\alpha_B$. Note that for $\alpha = 1$, this recovers the competitiveness obtained by Feldman et al. (2009a).

*Proof.* Consider an iteration where we assign an impression $t$ to advertiser $a$ and let $w_1 \leq w_2 \leq \cdots \leq w_{B_a}$ be the values of impressions currently allocated to $a$ in non-decreasing order. Let $w_0$ be the least valuable of the impressions allocated to $a$ at the end of iteration $t-1$, i.e. the impression that is removed to make space for $t$. Assume that after allocating impression $t$ to $a$, it becomes the $k$-th least valuable impression allocated to $a$ with value $w_{at} = w_k$. Thus, using that $w_i \geq w_{i-1}$, we can bound

$$\beta_a^{(t-1)} = \frac{e_{B_a}^{\alpha/B_a} - 1}{e_{B_a}^\alpha - 1}\left(\sum_{i=0}^{k-1} w_i e_{B_a}^{\alpha(B_a-i-1)/B_a} + \sum_{i=k+1}^{B_a} w_i e_{B_a}^{\alpha(B_a-i)/B_a}\right)$$

$$= \frac{e_{B_a}^{\alpha/B_a} - 1}{e_{B_a}^\alpha - 1}\left(\sum_{i=0}^{B_a-1} w_i e_{B_a}^{\alpha(B_a-i-1)/B_a} + \sum_{i=k+1}^{B_a}(w_i - w_{i-1})e_{B_a}^{\alpha(B_a-i)/B_a}\right)$$

$$\geq \frac{e_{B_a}^{\alpha/B_a} - 1}{e_{B_a}^\alpha - 1}\left(\sum_{i=0}^{B_a-1} w_i e_{B_a}^{\alpha(B_a-i-1)/B_a}\right) =: \hat{\beta}_a^{(t-1)}$$

which is tight when impression $t$ becomes the most valuable impression assigned to $a$. We can now write $\beta_a^{(t)}$ as a function of the bound $\hat{\beta}_a^{(t-1)}$:

$$\beta_a^{(t)} = \frac{e_{B_a}^{\alpha/B_a} - 1}{e_{B_a}^\alpha - 1}\sum_{i=1}^{B_a} w_i e_{B_a}^{\alpha(B_a-i)/B_a}$$

$$= \frac{e_{B_a}^{\alpha/B_a} - 1}{e_{B_a}^{\alpha} - 1} \left( \sum_{i=0}^{B_a-1} w_i e_{B_a}^{\alpha(B_a-i)/B_a} + w_{B_a} - w_0 e_{B_a}^{\alpha} \right)$$

$$= \frac{e_{B_a}^{\alpha/B_a} - 1}{e_{B_a}^{\alpha} - 1} e_{B_a}^{\alpha/B_a} \sum_{i=1}^{B_a} w_{i-1} e_{B_a}^{\alpha(B_a-i)/B_a} + \frac{e_{B_a}^{\alpha/B_a} - 1}{e_{B_a}^{\alpha} - 1} \left( w_{B_a} - w_0 e_{B_a}^{\alpha} \right)$$

$$= e_{B_a}^{\alpha/B_a} \hat{\beta}_a^{(t-1)} + \frac{e_{B_a}^{\alpha/B_a} - 1}{e_{B_a}^{\alpha} - 1} \left( w_{B_a} - w_0 e_{B_a}^{\alpha} \right).$$

We set $z_t := \alpha_B \left( w_{B_a} - \beta_a^{(t-1)} \right)$ which is feasible as the discounted value $w_{at} - \beta_a^{(t-1)}$ of the chosen advertiser $a$ may only be $\alpha_B$-times less the maximum discounted value $w_{a_{(\mathrm{EXP})}t} - \beta_{a_{(\mathrm{EXP})}}^{(t-1)}$ due to the advantage of the predicted advertiser. This yields a dual increase of

$$\Delta D = B_a \left( \beta_a^{(t)} - \beta_a^{(t-1)} \right) + z_t$$

$$= B_a \left( \beta_a^{(t)} - \beta_a^{(t-1)} \right) + \alpha_B \left( w_{B_a} - \beta_a^{(t-1)} \right)$$

$$\leq B_a \left( \beta_a^{(t)} - \hat{\beta}_a^{(t-1)} \right) + \alpha_B \left( w_{B_a} - \hat{\beta}_a^{(t-1)} \right)$$

$$= B_a \left( \left( e_{B_a}^{\alpha/B_a} - 1 \right) \hat{\beta}_a^{(t-1)} + \frac{e_{B_a}^{\alpha/B_a} - 1}{e_{B_a}^{\alpha} - 1} \left( w_{B_a} - w_0 e_{B_a}^{\alpha} \right) \right) + \alpha_B \left( w_{B_a} - \hat{\beta}_a^{(t-1)} \right)$$

$$= \alpha_{B_a} \hat{\beta}_a^{(t-1)} + \frac{\alpha_{B_a}}{e_{B_a}^{\alpha} - 1} \left( w_{B_a} - w_0 e_{B_a}^{\alpha} \right) + \alpha_B \left( w_{B_a} - \hat{\beta}_a^{(t-1)} \right)$$

$$= \alpha_{B_a} \frac{e_{B_a}^{\alpha}}{e_{B_a}^{\alpha} - 1} \underbrace{\left( \hat{\beta}_a^{(t-1)} - w_0 \right)}_{\geq 0} + \frac{\alpha_{B_a}}{e_{B_a}^{\alpha} - 1} \underbrace{\left( w_{B_a} - \hat{\beta}_a^{(t-1)} \right)}_{\geq 0} + \alpha_B \left( w_{B_a} - \hat{\beta}_a^{(t-1)} \right)$$

$$\leq \alpha_B \frac{e_B^{\alpha}}{e_B^{\alpha} - 1} \left( \hat{\beta}_a^{(t-1)} - w_0 \right) + \frac{\alpha_B}{e_B^{\alpha} - 1} \left( w_{B_a} - \hat{\beta}_a^{(t-1)} \right) + \alpha_B \left( w_{B_a} - \hat{\beta}_a^{(t-1)} \right)$$

$$= \alpha_B \frac{e_B^{\alpha}}{e_B^{\alpha} - 1} \left( \hat{\beta}_a^{(t-1)} - w_0 \right) + \alpha_B \frac{e_B^{\alpha}}{e_B^{\alpha} - 1} \left( w_{B_a} - \hat{\beta}_a^{(t-1)} \right)$$

$$= \alpha_B \frac{e_B^{\alpha}}{e_B^{\alpha} - 1} \left( w_{B_a} - w_0 \right) = B \frac{e_B^{\alpha/B} - 1}{e_B^{\alpha} - 1} e_B^{\alpha} \left( w_{B_a} - w_0 \right)$$

where the second inequality is due to $\alpha_B \geq \alpha_{B_a}$ and $e_B \leq e_{B_a}$, as shown in Lemma 2. $\qquad \square$

### A.2   Proof of Theorem 1 (Consistency)

In the following, we upper bound PRD using the comparison in Line 7 of Algorithm 1.

**Lemma 3.** *We have*

$$\mathrm{PRD} \leq \sum_a \left( (B_a - \ell_a) \beta_a^{(T)} + \frac{1}{\alpha_B} \sum_{t \in \mathbf{X}_a \setminus \mathbf{P}_a} \left( w_{at} - \beta_a^{(t-1)} \right) + \sum_{t \in \mathbf{P}_a \cap \mathbf{X}_a} w_{at} \right)$$

*Proof.* We first split impressions $t$ into two categories: Either the algorithm followed the prediction and assigned $t$ to $a^{(t)} = a_{(\mathrm{PRD})}^{(t)}$, or the algorithm ignored the prediction and assigned $t$ to $a^{(t)} = a_{(\mathrm{EXP})}^{(t)} \neq a_{(\mathrm{PRD})}^{(t)}$. In the latter case, due to the selection rule in Line 7 of Algorithm 1,

$$\alpha_B \left( w_{a_{(\mathrm{PRD})}^{(t)} t} - \beta_{a_{(\mathrm{PRD})}^{(t)}}^{(t-1)} \right) \leq w_{a_{(\mathrm{EXP})}^{(t)} t} - \beta_{a_{(\mathrm{EXP})}^{(t)}}^{(t-1)}.$$

In symbols,

$$\mathrm{PRD} = \sum_a \left( \sum_{t \in \mathbf{P}_a \setminus \mathbf{X}_a} w_{at} + \sum_{t \in \mathbf{P}_a \cap \mathbf{X}_a} w_{at} \right)$$

$$\leq \sum_a \left( \sum_{t \in \mathbf{P}_a \setminus \mathbf{X}_a} \left( \beta_a^{(t-1)} + \frac{1}{\alpha_B} \left( w_{a_{(\text{EXP})}^{(t)},t} - \beta_{a_{(\text{EXP})}^{(t)}}^{(t-1)} \right) \right) + \sum_{t \in \mathbf{P}_a \cap \mathbf{X}_a} w_{at} \right)$$

$$= \sum_a \left( \underbrace{\sum_{t \in \mathbf{P}_a \setminus \mathbf{X}_a} \beta_a^{(t-1)}}_{(\dagger)} + \frac{1}{\alpha_B} \sum_{t \in \mathbf{X}_a \setminus \mathbf{P}_a} \left( w_{at} - \beta_a^{(t-1)} \right) + \sum_{t \in \mathbf{P}_a \cap \mathbf{X}_a} w_{at} \right)$$

where the last equality holds because $\{\mathbf{P}_a\}_a$ and $\{\mathbf{X}_a\}_a$ are both partitioning the set of all impressions due to the introduction of the dummy advertiser. For $(\dagger)$, we use that $\beta_a$ can only increase in each round and bound

$$\sum_{t \in \mathbf{P}_a \setminus \mathbf{X}_a} \beta_a^{(t-1)} \leq (B_a - \ell_a) \beta_a^{(T)}.$$

$\square$

For the remainder of this section, we consider a fixed advertiser $a$. Let us denote with $t_i$ the $i$-th impression allocated to $a$. Let

$$(\star) = \frac{1}{\alpha_B} \sum_{t \in \mathbf{X}_a \setminus \mathbf{P}_a} \left( w_{at} - \beta_a^{(t-1)} \right) + \sum_{t \in \mathbf{P}_a \cap \mathbf{X}_a} w_{at}$$

as part of the the bound on PRD in Lemma 3.

In order to understand this bound, we make some useful observations in the following lemma to simplify the analysis. The key idea is that we may assume that impressions in $\mathbf{X}_a$ are ordered to be non-decreasing. In particular, we need to argue that the sum $\sum_{t \in \mathbf{X}_a \setminus \mathbf{P}_a} \beta_a^{(t-1)}$ can only decrease (as this term is negated in $(\star)$) when impressions in $\mathbf{X}_a$ are ordered to be non-decreasing: Intuitively, each $\beta_a^{(t-1)}$ depends only on the $B_a$ most valuable impressions assigned before impression $t$, no matter the order in which $\mathbf{X}_a^{(t-1)}$ arrived. We can thus minimize each $\beta_a^{(t-1)}$ if the impressions allocated prior to $t$ are the impressions of smallest value. To simultaneously minimize each $\beta_a^{(t)}$ in the sum, we order the impressions in $\mathbf{X}_a$ to have non-decreasing value. We prove this simplification formally in the following lemma.

**Lemma 4.** *Without loss of generality, we may assume that $\mathbf{P}_a \cap \mathbf{X}_a$ are the most valuable impressions in $\mathbf{X}_a$ and that impressions in $\mathbf{X}_a$ arrive such that their values are non-decreasing.*

*Proof.* We may assume that the impressions in $\mathbf{P}_a \cap \mathbf{X}_a$ are the most valuable impressions in $\mathbf{X}_a$: this can only increase the value of $\mathbf{P}_a$ but leaves $\mathbf{S}_a$ unaffected, as $\mathbf{S}_a$ are by design the $B_a$ most valuable impressions in $\mathbf{X}_a$. All impressions in $(\star)$ are from $\mathbf{X}_a$, so reordering impressions only affects $(\star)$. Specifically, we can show that the sum $\sum_{t \in \mathbf{X}_a \setminus \mathbf{P}_a} \beta_a^{(t-1)}$ in $(\star)$ is minimized if the values in $\mathbf{X}_a$ are ordered to be non-decreasing. Assume to the contrary that the $i$-th impression added to $a$ is the last that is in order. That is $w_{at_1} \leq w_{at_2} \leq \cdots \leq w_{at_i}$ and there exists a $j \leq i$ such that $w_{at_{j-1}} \leq w_{at_{i+1}} < w_{at_j}$. Moving $t_{i+1}$ ahead to its ranked position within the first $i$ impressions allocated to $a$ changes the ordering as follows (the first and second row show the impression values before and after changing the position of $t_{i+1}$, respectively):

$$w_{at_1} \leq \cdots \leq w_{at_{j-1}} \leq \ w_{at_j} \ \leq w_{at_{j+1}} \leq \cdots \leq \ w_{at_i}$$
$$w_{at_1} \leq \cdots \leq w_{at_{j-1}} \leq w_{at_{i+1}} < \ w_{at_j} \ \leq \cdots \leq w_{at_{i-1}}$$

Note that each position decreases in value, even strictly at the $j$-th position. As such, the exponential average $\beta_a^{(t-1)}$ decreases as well for $t < t_{i+1}$; it remains constant for $t \geq t_{i+1}$ as it only depends on the $B_a$ most valuable impressions assigned up to $t$ which remain the same. We can thus simultaneously minimize $\beta_a^{(t)}$ for each $t$ by putting $\mathbf{X}_a$ in non-decreasing order. This reordering does not affect $\beta_a^{(T)}$ or the other terms in $(\star)$, so we may indeed assume that values are non-decreasing. $\square$

In light of Lemma 4, we can write $(\star)$ as follows.

**Lemma 5.** *We have*

$$(\star) = \frac{1}{\alpha_B} \sum_{i=1}^{I_a - \ell_a} \left( w_{at_i} - \beta_a^{(t_{i-1})} \right) + \sum_{i=I_a - \ell_a + 1}^{I_a} w_{at_i}.$$

*Proof.* Impression values are non-decreasing due to Lemma 4, so $w_{at_i}$ is the $i$-th least valuable impression in $\mathbf{X}_a$. The impressions $\{I_a - \ell_a + 1, \ldots, I_a\} = \mathbf{X}_a \cap \mathbf{P}_a$ are thus the most valuable. We can now write $(\star)$ as

$$\frac{1}{\alpha_B} \sum_{t \in \mathbf{X}_a \setminus \mathbf{P}_a} \left( w_{at} - \beta_a^{(t-1)} \right) + \sum_{t \in \mathbf{P}_a \cap \mathbf{X}_a} w_{at} = \frac{1}{\alpha_B} \sum_{i=1}^{I_a - \ell_a} \left( w_{at_i} - \beta_a^{(t_{i-1})} \right) + \sum_{i=I_a - \ell_a + 1}^{I_a} w_{at_i}$$

where $\beta_a^{(t_i - 1)} = \beta_a^{(t_{i-1})}$ as there was no change to the dual variable of advertiser $a$ since no impression in $\{t_{i-1} + 1, \ldots, t_i - 1\}$ was allocated to $a$. $\qquad\square$

Combining Lemmas 3 and 5, we obtain:

**Lemma 6.** $\mathrm{PRD} \leq \sum_a \mathrm{PRD}_a$ *where*

$$\mathrm{PRD}_a := \frac{1}{\alpha_B} \sum_{i=1}^{I_a - \ell_a} \left( w_{at_i} - \beta_a^{(t_{i-1})} \right) + \sum_{i=I_a - \ell_a + 1}^{I_a} w_{at_i} + (I_a - \ell_a) \beta_a^{(T)}.$$

In the following, we use the non-decreasing ordering of impressions in $\mathbf{X}_a$ to compute $\beta_a^{(t_{i-1})}$ and bound $\mathrm{PRD}_a$ with a linear combination of values $w_{at_i}$. Consider the $j$-th impression $t_j$ allocated to $a$. Since we assume that impression values are non-decreasing, we know that $t_j$ becomes the most valuable impression right after it is allocated. After the allocation of the $(j + 1)$-th impression to $a$, it becomes the second most valuable impression, and so forth, until it is disposed after the allocation of the $(j + B_a)$-th impression. The value $w_{at_j}$ therefore appears alongside each coefficient in the convex combination that defines $\beta_a^{(t_{i-1})}$ for $i \in \{j + 1, \ldots, j + B_a\}$. Expanding each $\beta_a^{(t_{i-1})}$ in the sum $\sum_{i=1}^{I_a - \ell_a} \beta_a^{(t_{i-1})}$ in $\mathrm{PRD}_a$, we thus observe that the coefficients of values $w_{at_j}$ for $j \leq I_a - \ell_a - B_a$ sum up to 1. We use this fact to cancel out most of the values in $\sum_{i=1}^{I_a - \ell_a} w_{at_i}$. What remains are only the values $w_{at_i}$ for $i \in \{I_a - \ell_a - B_a + 1, \ldots, I_a - \ell_a\}$. For $i \in \{I_a - \ell_a - B_a + 1, \ldots, I_a - B_a\}$, we bound $w_{at_i}$ by $w_{at_{I_a - B_a}}$ which is really the best we can hope for. Formally, we show:

**Lemma 7.** *We have*

$$\mathrm{PRD}_a \leq \sum_{i=I_a - B_a + 1}^{I_a - \ell_a} \phi_i w_{at_i} + \sum_{i=I_a - \ell_a + 1}^{I_a} \psi_i w_{at_i} + w_{at_{I_a - B_a}} \Omega_a$$

*with coefficients*

$$\phi_i := (B_a - \ell_a) \frac{e_{B_a}^{\alpha/B_a} - 1}{e_{B_a}^{\alpha} - 1} e_{B_a}^{\alpha(I_a - i)/B_a} + \frac{1}{\alpha_B} \frac{e_{B_a}^{\alpha} - e_{B_a}^{\alpha(I_a - \ell_a - i)/B_a}}{e_{B_a}^{\alpha} - 1}$$

$$\psi_i := 1 + (B_a - \ell_a) \frac{e_{B_a}^{\alpha/B_a} - 1}{e_{B_a}^{\alpha} - 1} e_{B_a}^{\alpha(I_a - i)/B_a}$$

$$\Omega_a := \frac{1}{\alpha_B} \frac{1}{e_{B_a}^{\alpha} - 1} \left( \ell_a e_{B_a}^{\alpha} - \frac{e_{B_a}^{\alpha} - e_{B_a}^{\alpha(B_a - \ell_a)/B_a}}{e_{B_a}^{\alpha/B_a} - 1} \right).$$

*Proof.* We start by rewriting the terms in $\mathrm{PRD}_a$ individually. Since we assume that the values are non-decreasing, we can express $\beta_a^{(t_{i-1})}$ as the exponential average of values $w_{at_{i - B_a}}, w_{at_{i - B_a + 1}}, \ldots, w_{at_{i-1}}$ of the last $B_a$ impressions (for simplicity, we set $w_{at_j} = 0$ for $j \leq 0$). Summing over multiple iterations, we thus obtain for the sum over the dual variables that

$$\sum_{i=1}^{I_a - \ell_a} \beta_a^{(t_{i-1})} = \frac{e_{B_a}^{\alpha/B_a} - 1}{e_{B_a}^{\alpha} - 1} \sum_{i=1}^{I_a - \ell_a} \sum_{j=i - B_a}^{i-1} w_{at_j} e_{B_a}^{\alpha(i - j - 1)/B_a}$$

$$= \frac{e_{B_a}^{\alpha/B_a} - 1}{e_{B_a}^{\alpha} - 1} \sum_{j=1}^{I_a - \ell_a} w_{at_j} \sum_{i=j+1}^{\min\{j+B_a, I_a-\ell_a\}} e_{B_a}^{\alpha(i-j-1)/B_a}$$

$$= \frac{e_{B_a}^{\alpha/B_a} - 1}{e_{B_a}^{\alpha} - 1} \sum_{j=1}^{I_a - \ell_a} w_{at_j} \sum_{i=1}^{\min\{B_a, I_a-\ell_a-j\}} e_{B_a}^{\alpha(i-1)/B_a}$$

$$= \frac{e_{B_a}^{\alpha/B_a} - 1}{e_{B_a}^{\alpha} - 1} \sum_{j=1}^{I_a - B_a - \ell_a} w_{at_j} \sum_{i=1}^{I_a} e_{B_a}^{\alpha(i-1)/B_a}$$

$$+ \frac{e_{B_a}^{\alpha/B_a} - 1}{e_{B_a}^{\alpha} - 1} \sum_{j=I_a-B_a-\ell_a+1}^{I_a - \ell_a} w_a \sum_{i=1}^{I_a-\ell_a-j} e_{B_a}^{\alpha(i-1)/B_a}$$

$$= \sum_{i=1}^{I_a - B_a - \ell_a} w_{at_i} + \frac{1}{e_{B_a}^{\alpha} - 1} \sum_{i=I_a-B_a-\ell_a+1}^{I_a - \ell_a} w_{at_i} \left( e_{B_a}^{\alpha(I_a-\ell_a-i)/B_a} - 1 \right).$$

where for the last equality, we use that the two inner sums are geometric. We can use this expression to cancel out most of the terms of the first sum in $\mathrm{PRD}_a$:

$$\sum_{i=1}^{I_a - \ell_a} \left( w_{at_i} - \beta_a^{(t_{i-1})} \right)$$

$$= \sum_{i=1}^{I_a - \ell_a} w_{at_i} - \sum_{i=1}^{I_a - B_a - \ell_a} w_{at_i} - \frac{1}{\alpha_B \left( e_{B_a}^{\alpha} - 1 \right)} \sum_{i=I_a-B_a-\ell_a+1}^{I_a - \ell_a} w_{at_i} \left( e_{B_a}^{\alpha(I_a-\ell_a-i)/B_a} - 1 \right)$$

$$= \sum_{i=I_a-B_a-\ell_a+1}^{I_a - \ell_a} w_{at_i} \left( 1 - \frac{e_{B_a}^{\alpha(I_a-\ell_a-i)/B_a} - 1}{e_{B_a}^{\alpha} - 1} \right)$$

$$= \sum_{i=I_a-B_a-\ell_a+1}^{I_a - \ell_a} w_{at_i} \frac{e_{B_a}^{\alpha} - e_{B_a}^{\alpha(I_a-\ell_a-i)/B_a}}{e_{B_a}^{\alpha} - 1}$$

$$= \sum_{i=I_a-B_a+1}^{I_a - \ell_a} w_{at_i} \frac{e_{B_a}^{\alpha} - e_{B_a}^{\alpha(I_a-\ell_a-i)/B_a}}{e_{B_a}^{\alpha} - 1} + \sum_{i=I_a-B_a-\ell_a+1}^{I_a - B_a} w_{at_i} \frac{e_{B_a}^{\alpha} - e_{B_a}^{\alpha(I_a-\ell_a-i)/B_a}}{e_{B_a}^{\alpha} - 1}. \qquad (3)$$

We use that $w_{at_i} \leq w_{at_{I_a - B_a}}$ for all $i \leq I_a - B_a$ to upper bound the second sum, divided by $\alpha_B$, in (3) to

$$\frac{1}{\alpha_B} \sum_{i=I_a-B_a-\ell_a+1}^{I_a - B_a} w_{at_i} \frac{e_{B_a}^{\alpha} - e_{B_a}^{\alpha(I_a-\ell_a-i)/B_a}}{e_{B_a}^{\alpha} - 1}$$

$$\leq w_{at_{I_a - B_a}} \frac{1}{\alpha_B} \sum_{i=I_a-B_a-\ell_a+1}^{I_a - B_a} \frac{e_{B_a}^{\alpha} - e_{B_a}^{\alpha(I_a-\ell_a-i)/B_a}}{e_{B_a}^{\alpha} - 1} \qquad (4)$$

$$= w_{at_{I_a - B_a}} \frac{1}{\alpha_B} \frac{1}{e_{B_a}^{\alpha} - 1} \left( \ell_a e_{B_a}^{\alpha} - \sum_{i=B_a-\ell_a}^{B_a-1} e_{B_a}^{\alpha i/B_a} \right)$$

$$= w_{at_{I_a - B_a}} \underbrace{\frac{1}{\alpha_B} \frac{1}{e_{B_a}^{\alpha} - 1} \left( \ell_a e_{B_a}^{\alpha} - \frac{e_{B_a}^{\alpha} - e_{B_a}^{\alpha(B_a-\ell_a)/B_a}}{e_{B_a}^{\alpha/B_a} - 1} \right)}_{=\Omega_a}. \qquad (5)$$

Furthermore, by definition of $\beta_a^{(T)} = \beta_a^{(t_{I_a})}$,

$$(B_a - \ell_a) \beta_a^{(T)} = (B_a - \ell_a) \frac{e_{B_a}^{\alpha/B_a} - 1}{e_{B_a}^{\alpha} - 1} \sum_{i=I_a-B_a+1}^{I_a} w_{at_i} e_{B_a}^{\alpha(I_a-i)/B_a}. \qquad (6)$$

We combine (3), (5), and (6) and group terms to obtain the desired bound

$$\mathrm{PRD}_a \leq \sum_{i=I_a-\ell_a+1}^{I_a} w_{at_i} + \frac{1}{\alpha_B} \sum_{i=I_a-B_a+1}^{I_a-\ell_a} w_{at_i} \frac{e_{B_a}^{\alpha} - e_{B_a}^{\alpha(I_a-\ell_a-i)/B_a}}{e_{B_a}^{\alpha} - 1} + w_{at_{I_a-B_a}} \Omega_a$$

$$+ (B_a - \ell_a) \frac{e_{B_a}^{\alpha/B_a} - 1}{e_{B_a}^{\alpha} - 1} \sum_{i=I_a-B_a+1}^{I_a} w_{at_i} e_{B_a}^{\alpha(I_a-i)/B_a}$$

$$= \sum_{i=I_a-B_a+1}^{I_a-\ell_a} w_{at_i} \underbrace{\left( (B_a - \ell_a) \frac{e_{B_a}^{\alpha/B_a} - 1}{e_{B_a}^{\alpha} - 1} e_{B_a}^{\alpha(I_a-i)/B_a} + \frac{1}{\alpha_B} \frac{e_{B_a}^{\alpha} - e_{B_a}^{\alpha(I_a-\ell_a-i)/B_a}}{e_{B_a}^{\alpha} - 1} \right)}_{=\phi_i}$$

$$+ \sum_{i=I_a-\ell_a+1}^{I_a} w_{at_i} \underbrace{\left( 1 + (B_a - \ell_a) \frac{e_{B_a}^{\alpha/B_a} - 1}{e_{B_a}^{\alpha} - 1} e_{B_a}^{\alpha(I_a-i)/B_a} \right)}_{=\psi_i} + w_{I_a-B_a} \Omega_a$$

$\square$

We can express ALG analogously:

**Lemma 8.** *We have* $\mathrm{ALG} = \sum_a \mathrm{ALG}_a$ *where*

$$\mathrm{ALG}_a := \sum_{i=I_a-B_a+1}^{I_a} w_{at_i}.$$

*Proof.* We have $\mathrm{ALG} = \sum_a \sum_{t \in \mathbf{S}_a} w_{at}$. As we always dispose of the least valuable impression in Algorithm 1, $\mathbf{S}_a$ are the $B_a$ most valuable impressions in $\mathbf{X}_a$. Due to Lemma 4, these are $\mathbf{S}_a = \{I_a - B_a + 1, \ldots, I_a\}$ and hence $\sum_{t \in \mathbf{S}_a} w_{at} = \sum_{i=I_a-B_a+1}^{I_a} w_{at_i} = \mathrm{ALG}_a$. $\square$

We upper bound the ratio $\mathrm{PRD}/\mathrm{ALG}$ by $\max_a \mathrm{PRD}_a/\mathrm{ALG}_a$. To this end, we fix an advertiser $a$ and upper bound the ratio $\mathrm{PRD}_a/\mathrm{ALG}_a$. Recall from Lemmas 7 and 8 that we can express $\mathrm{PRD}_a$ and $\mathrm{ALG}_a$ as linear combination over impression values. We could obtain a natural upper bound by comparing impression value coefficients. However, in the following lemma, we show how to use the non-decreasing ordering due to Lemma 4 to obtain a tighter bound.

We define

$$\Phi_a := \sum_{i=I_a-B_a+1}^{I_a-\ell_a} \phi_i \qquad \text{and} \qquad \Psi_a := \sum_{i=I_a-\ell_a+1}^{I_a} \psi_i$$

as the total factor mass on values $w_{at_i}$ for $\phi_i$ and $\psi_i$, respectively. Let $\tau_a := (\Phi_a + \Psi_a + \Omega_a)/B_a$ be the average factor. Recall that

$$\mathrm{PRD}_a \leq \sum_{i=I_a-B_a+1}^{I_a-\ell_a} \phi_i w_{at_i} + \sum_{i=I_a-\ell_a+1}^{I_a} \psi_i w_{at_i} + w_{at_{I_a-B_a}} \Omega_a \qquad (7)$$

$$\mathrm{ALG}_a = \sum_{i=I_a-B_a+1}^{I_a} w_{at_i}.$$

In the following lemma, we use that $w_{at_i} \leq w_{at_j}$ for $i \leq j$ due to Lemma 4, to further upper bound the RHS of 7 by a linear combination of the values, where we move mass from coefficients on $w_{at_i}$ to coefficients on $w_{at_j}$. Additionally, we move mass from $\Omega_a$ to coefficients $\phi_i$ for $i \in \{I_a - B_a + 1, \ldots, I_a - \ell_a\}$ and from $\phi_i$ to $\psi_j$ for $j \in \{I_a - \ell_a + 1, \ldots, I_a\}$. In the best case, we are able to redistribute mass equally across all values, in which case the consistency is given as the average factor $\tau_a$. Otherwise, the factors on the largest values dominate, giving us a consistency of $\Psi_a/\ell_a$.

**Lemma 9.** *We have*

$$\frac{\mathrm{PRD}_a}{\mathrm{ALG}_a} \leq \begin{cases} \max\left\{\tau_a, \frac{\Psi_a}{\ell_a}\right\} & \text{if } \ell_a > 0 \\ \tau_a & \text{otherwise} \end{cases}$$

*where*

$$\tau_a = 1 + \frac{1}{e_{B_a}^{\alpha} - 1} \frac{1}{\alpha_B}\left(e_{B_a}^{\alpha} - \frac{e_{B_a}^{\alpha} - 1}{\alpha_{B_a}}\right)$$

*and*

$$\frac{\Psi_a}{\ell_a} = 1 + \left(\frac{B_a}{\ell_a} - 1\right)\frac{e_{B_a}^{\alpha\ell_a/B_a} - 1}{e_{B_a}^{\alpha} - 1}.$$

*Proof.* We calculate $\tau_a$ and $\Psi_a/\ell_a$ separately in Lemma 10 below. Our main goal is to distribute mass from the factors $\phi_i$, $\psi_i$, and from $\Omega_a$ equally to the values $w_{I_a - B_a + 1}, \ldots, w_{I_a}$. We begin by taking a closer look at the factors $\phi_i$ and $\psi_i$. First, note that $\psi_i$ is always decreasing in $i$ as

$$\psi_i = 1 + \underbrace{(B_a - \ell_a)}_{\geq 0} \underbrace{\frac{e_{B_a}^{\alpha/B_a} - 1}{e_{B_a}^{\alpha} - 1}}_{\geq 0} e_{B_a}^{\alpha(I_a - i)/B_a}.$$

We can therefore bound the linear combination over values in $\{I_a - \ell_a + 1, \ldots, I_a\}$ using the average value $\bar{w}_\Psi := \frac{1}{\ell_a}\sum_{i=I_a-\ell_a+1}^{I_a} w_{at_i}$ as

$$\sum_{i=I_a-\ell_a+1}^{I_a} w_{at_i}\psi_i \leq \sum_{i=I_a-\ell_a+1}^{I_a} \bar{w}_\Psi \psi_i = \bar{w}_\Psi \Psi_a. \tag{8}$$

However, $\phi_i$ is not always decreasing which can be seen by rearranging

$$\phi_i = (B_a - \ell_a)\frac{e_{B_a}^{\alpha/B_a} - 1}{e_{B_a}^{\alpha} - 1}e^{\alpha(I_a-i)/B_a} + \frac{1}{\alpha_B}\frac{e_{B_a}^{\alpha} - e_{B_a}^{\alpha(I_a-\ell_a-i)/B_a}}{e_{B_a}^{\alpha} - 1}$$

$$= \frac{1}{e_{B_a}^{\alpha} - 1}\left((B_a - \ell_a)\left(e_{B_a}^{\alpha/B_a} - 1\right) - \frac{1}{\alpha_B}e_{B_a}^{-\alpha\ell_a/B_a}\right)e_{B_a}^{\alpha(I_a-i)/B_a} + \frac{1}{\alpha_B}\frac{e_{B_a}^{\alpha}}{e_{B_a}^{\alpha} - 1}.$$

We observe that $\phi_i$ is decreasing if $(B_a - \ell_a)\left(e_{B_a}^{\alpha/B_a} - 1\right)$ is at least $\frac{1}{\alpha_B}e_{B_a}^{-\alpha\ell_a/B_a}$, and we analyze two cases based on the relationship of both terms.

Let us first assume that $(B_a - \ell_a)\left(e_{B_a}^{\alpha/B_a} - 1\right) \geq \frac{1}{\alpha_B}e_{B_a}^{-\alpha\ell_a/B_a}$ such that $\phi_i$ is decreasing in $i$ which helps us to bound the linear combinations in Lemma 7 over $\{I_a - B_a + 1, \ldots, I_a - \ell_a\}$ and $\{I_a - \ell_a + 1, \ldots, T\}$ by the average values $\bar{w}_\Phi := \frac{1}{B_a - \ell_a}\sum_{i=I_a-B_a+1}^{I_a-\ell_a} w_{at_i}$ and $\bar{w}_\Psi$, respectively. We further use that $w_{at_{I_a - B_a}} \leq \bar{w}_\Phi$ to charge mass from $\Omega_a$ to $\Phi_a$ and obtain due to (8) that

$$\sum_{i=I_a-B_a+1}^{I_a-\ell_a} w_{at_i}\phi_i + \sum_{i=I_a-\ell_a+1}^{I_a} w_{at_i}\psi_i + w_{at_{I_a-B_a}}\Omega_a$$

$$\leq \bar{w}_\Phi \Phi_a + \bar{w}_\Psi \Psi_a + w_{at_{I_a-B_a}}\Omega_a \tag{9}$$

$$\leq \bar{w}_\Phi (\Phi_a + \Omega_a) + \bar{w}_\Psi \Psi_a$$

$$= \bar{w}_\Phi (B_a - \ell_a)\tau_a + \bar{w}_\Phi (\Phi_a + \Omega_a - (B_a - \ell_a)\tau_a) + \bar{w}_\Psi \Psi_a$$

$$= \sum_{i=I_a-B_a+1}^{I_a-\ell_a} \tau_a w_{at_i} + \bar{w}_\Phi (\Phi_a + \Omega_a - (B_a - \ell_a)\tau_a) + \bar{w}_\Psi \Psi_a \tag{10}$$

On the other hand, if $(B_a - \ell_a)\left(e_{B_a}^{\alpha/B_a} - 1\right) \leq \frac{1}{\alpha_B}e_{B_a}^{-\alpha\ell_a/B_a}$ we can no longer bound the values over $\{I_a - B_a + 1, \ldots, I_a - \ell_a\}$ by the average value $\bar{w}_\Phi$. However, each factor $\phi_i$ is less than $\tau_a$ which can be seen by rearranging

$$\phi_i = \frac{1}{e_{B_a}^{\alpha} - 1}\left((B_a - \ell_a)\left(e_{B_a}^{\alpha/B_a} - 1\right) - \frac{1}{\alpha_B}e_{B_a}^{-\alpha\ell_a/B_a}\right)e_{B_a}^{\alpha(I_a-i)/B_a} + \frac{1}{\alpha_B}\frac{e_{B_a}^{\alpha}}{e_{B_a}^{\alpha} - 1}$$

$$\leq 1 + \frac{1}{e_{B_a}^{\alpha} - 1}\frac{1}{\alpha_B}\left(e_{B_a}^{\alpha} - \frac{e_{B_a}^{\alpha} - 1}{\alpha_{B_a}}\right) = \tau_a$$

to the equivalent expression

$$\underbrace{\left((B_a - \ell_a)\left(e_{B_a}^{\alpha/B_a} - 1\right) - \frac{1}{\alpha_B}e_{B_a}^{-\alpha\ell_a/B_a}\right)}_{\leq 0}\underbrace{e_{B_a}^{\alpha(I_a-i)/B_a}}_{\geq 0}$$

$$\leq e_{B_a}^{\alpha} - 1 - \frac{1}{\alpha_B}\frac{e_{B_a}^{\alpha} - 1}{\alpha_{B_a}} = \underbrace{\left(1 - \frac{1}{\alpha_B \cdot a_{B_a}}\right)}_{\geq 0}\underbrace{\left(e_{B_a}^{\alpha} - 1\right)}_{\geq 0}$$

which is true since the LHS is $\leq 0$ and the RHS $\geq 0$. We can thus charge $\tau_a - \phi_i$ of mass from $\Omega_a$ to the coefficients $\phi_i$ for each $i \in \{I_a - B_a + 1, \ldots, I_a - \ell_a\}$ which yields

$$\sum_{i=I_a-B_a+1}^{I_a-\ell_a} w_{at_i}\phi_i + \sum_{i=I_a-\ell_a+1}^{I_a} w_{at_i}\psi_i + w_{at_{I_a-B_a}}\Omega_a$$

$$\leq \sum_{i=I_a-B_a+1}^{I_a-\ell_a} w_{at_i}\phi_i + \bar{w}_\Psi\Psi_a + w_{at_{I_a-B_a}}\Omega_a$$

$$= \sum_{i=I_a-B_a+1}^{I_a-\ell_a} \tau_a w_{at_i} - \sum_{i=I_a-B_a+1}^{I_a-\ell_a} w_{at_i}\underbrace{(\tau_a - \phi_i)}_{\geq 0} + \bar{w}_\Psi\Psi_a + w_{at_{I_a-B_a}}\Omega_a$$

$$\leq \sum_{i=I_a-B_a+1}^{I_a-\ell_a} \tau_a w_{at_i} - \sum_{i=I_a-B_a+1}^{I_a-\ell_a} w_{at_{I_a-B_a}}(\tau_a - \phi_i) + \bar{w}_\Psi\Psi_a + w_{at_{I_a-B_a}}\Omega_a$$

$$= \sum_{i=I_a-B_a+1}^{I_a-\ell_a} \tau_a w_{at_i} + w_{at_{I_a-B_a}}\left(\Phi_a + \Omega_a - (B_a - \ell_a)\tau_a\right) + \bar{w}_\Psi\Psi_a \qquad (11)$$

In both cases (10) and (11), we have shown that

$$\sum_{i=I_a-B_a+1}^{I_a-\ell_a} w_{at_i}\phi_i + \sum_{i=I_a-\ell_a+1}^{I_a} w_{at_i}\psi_i + w_{at_{I_a-B_a}}\Omega_a$$

$$\leq \sum_{i=I_a-B_a+1}^{I_a-\ell_a} \tau_a w_{at_i} + v\left(\Phi_a + \Omega_a - (B_a - \ell_a)\tau_a\right) + \bar{w}_\Psi\Psi_a$$

for a $v \leq \bar{w}_\Psi$. If $\ell_a > 0$, we can use $v \leq \bar{w}_\Psi$ to charge the remaining mass to $\Psi_a$ if the factors over $\{I_a - \ell_a + 1, \ldots, T\}$ leave enough space. In symbols, this means

$$\sum_{i=T-B_a+1}^{I_a-\ell_a} \tau_a w_{at_i} + v\left(\Phi_a + \Omega_a - (B_a - \ell_a)\tau_a\right) + \bar{w}_\Psi\Psi_a$$

$$\leq \sum_{i=T-B_a+1}^{I_a-\ell_a} \tau_a w_{at_i} + \bar{w}_\Psi \max\left\{\Phi_a + \Omega_a - (B_a - \ell_a)\tau_a, 0\right\} + \bar{w}_\Psi\Psi_a$$

$$= \sum_{i=T-B_a+1}^{I_a-\ell_a} \tau_a w_{at_i} + \bar{w}_\Psi \max\left\{\Phi_a + \Psi_a + \Omega_a - (B_a - \ell_a)\tau_a, \Psi_a\right\}$$

$$= \sum_{i=T-B_a+1}^{I_a-\ell_a} \tau_a w_{at_i} + \bar{w}_\Psi \max\left\{\ell_a\tau_a, \Psi_a\right\}$$

$$\leq \tau_a \sum_{i=T-B_a+1}^{I_a-\ell_a} w_{at_i} + \max\left\{\tau_a, \frac{\Psi_a}{\ell_a}\right\} \sum_{i=T-\ell_a+1}^{I_a} w_{at_i}$$

$$\leq \max\left\{\tau_a, \frac{\Psi_a}{\ell_a}\right\} \sum_{t\in S_a} w_{at}.$$

If $\ell_a = 0$, we have $\Psi_a = 0$ and immediately obtain by definition of $\tau_a$ that

$$\sum_{i=I_a-B_a+1}^{I_a-\ell_a} \tau_a w_{at_i} + v\left(\Phi_a + \Omega_a - (B_a - \ell_a)\,\tau_a\right) + \bar{w}_\Psi \Psi_a = \sum_{i=I_a-B_a+1}^{I_a-\ell_a} \tau_a w_{at_i}.$$

$\square$

**Lemma 10.** *We have*
$$\frac{\Psi_a}{\ell_a} = 1 + \left(\frac{B_a}{\ell_a} - 1\right)\frac{e_{B_a}^{\alpha\ell_a/B_a} - 1}{e_{B_a}^\alpha - 1}$$

*and*
$$\tau_a = 1 + \frac{1}{e_{B_a}^\alpha - 1}\frac{1}{\alpha_B}\left(e_{B_a}^\alpha - \frac{e_{B_a}^\alpha - 1}{\alpha_{B_a}}\right).$$

*Proof.* We compute

$$\Phi_a = \sum_{i=T-B_a+1}^{T-\ell_a} \phi_i$$

$$= \sum_{i=T-B_a+1}^{T-\ell_a}\left(\frac{1}{e_{B_a}^\alpha - 1}\left((B_a - \ell_a)\left(e_{B_a}^{\alpha/B_a} - 1\right) - \frac{1}{\alpha_B}e_{B_a}^{-\alpha\ell_a/B_a}\right)e_{B_a}^{\alpha(T-i)/B_a} + \frac{1}{\alpha_B}\frac{e_{B_a}^\alpha}{e_{B_a}^\alpha - 1}\right)$$

$$= \frac{1}{e_{B_a}^\alpha - 1}\left((B_a - \ell_a)\left(e_{B_a}^{\alpha/B_a} - 1\right) - \frac{1}{\alpha_B}e_{B_a}^{-\alpha\ell_a/B_a}\right)\frac{e_{B_a}^\alpha - e_{B_a}^{\alpha\ell_a/B_a}}{e_{B_a}^{\alpha/B_a} - 1} + (B_a - \ell_a)\frac{1}{\alpha_B}\frac{e_{B_a}^\alpha}{e_{B_a}^\alpha - 1}$$

$$= \frac{1}{e_{B_a}^\alpha - 1}(B_a - \ell_a)\left(e_{B_a}^\alpha - e_{B_a}^{\alpha\ell_a/B_a} + \frac{1}{\alpha_B}e_{B_a}^\alpha\right) - \frac{1}{e_{B_a}^\alpha - 1}\frac{1}{\alpha_B}\frac{e_{B_a}^{\alpha-\alpha\ell_a/B_a} - 1}{e_{B_a}^{\alpha/B_a} - 1}$$

and

$$\Psi_a = \sum_{i=T-\ell_a+1}^{T} \psi_i$$

$$= \sum_{i=T-\ell_a+1}^{T}\left(1 + (B_a - \ell_a)\frac{e_{B_a}^{\alpha/B_a} - 1}{e_{B_a}^\alpha - 1}e_{B_a}^{\alpha(T-i)/B_a}\right)$$

$$= \ell_a + (B_a - \ell_a)\frac{e_{B_a}^{\alpha/B_a} - 1}{e_{B_a}^\alpha - 1}\frac{e_{B_a}^{\alpha\ell_a/B_a} - 1}{e_{B_a}^{\alpha/B_a} - 1}$$

$$= \ell_a + (B_a - \ell_a)\frac{e_{B_a}^{\alpha\ell_a/B_a} - 1}{e_{B_a}^\alpha - 1}.$$

Summing up,

$$\Phi_a + \Psi_a + \Omega_a$$

$$= \frac{1}{e^\alpha - 1}(B_a - \ell_a)\left(e_{B_a}^\alpha - e_{B_a}^{\alpha\ell_a/B_a} + \frac{1}{\alpha_B}e_{B_a}^\alpha\right) - \frac{1}{e_{B_a}^\alpha - 1}\frac{1}{\alpha_B}\frac{e_{B_a}^{\alpha-\alpha\ell_a/B_a} - 1}{e_{B_a}^{\alpha/B_a} - 1}$$

$$+ \ell_a + (B_a - \ell_a)\frac{e_{B_a}^{\alpha\ell_a/B_a} - 1}{e_{B_a}^\alpha - 1} + \frac{1}{e_{B_a}^\alpha - 1}\frac{1}{\alpha_B}\left(\ell_a e_{B_a}^\alpha - \frac{e_{B_a}^\alpha - e_{B_a}^{\alpha(B_a-\ell_a)/B_a}}{e_{B_a}^{\alpha/B_a} - 1}\right)$$

$$= \frac{1}{e^{\alpha}_{B_a} - 1} (B_a - \ell_a) \left( e^{\alpha}_{B_a} - 1 + \frac{1}{\alpha_B} e^{\alpha}_{B_a} \right) - \frac{1}{e^{\alpha}_{B_a} - 1} \frac{1}{\alpha_B} \frac{e^{\alpha}_{B_a} - 1}{e^{\alpha/B_a}_{B_a} - 1}$$

$$+ \ell_a + \frac{1}{e^{\alpha}_{B_a} - 1} \frac{1}{\alpha_B} \ell_a e^{\alpha}_{B_a}$$

$$= B_a + \frac{1}{e^{\alpha}_{B_a} - 1} \frac{1}{\alpha_B} (B_a - \ell_a) e^{\alpha}_{B_a} - \frac{1}{e^{\alpha}_{B_a} - 1} \frac{1}{\alpha_B} \frac{e^{\alpha}_{B_a} - 1}{e^{\alpha/B_a}_{B_a} - 1} + \frac{1}{e^{\alpha}_{B_a} - 1} \frac{1}{\alpha_B} \ell_a e^{\alpha}_{B_a}$$

$$= B_a + \frac{1}{e^{\alpha}_{B_a} - 1} \frac{1}{\alpha_B} B_a e^{\alpha}_{B_a} - \frac{1}{e^{\alpha}_{B_a} - 1} \frac{1}{\alpha_B} \frac{e^{\alpha}_{B_a} - 1}{e^{\alpha/B_a}_{B_a} - 1}$$

$$= B_a + \frac{1}{e^{\alpha}_{B_a} - 1} \frac{1}{\alpha_B} \left( B_a e^{\alpha}_{B_a} - \frac{e^{\alpha}_{B_a} - 1}{e^{\alpha/B_a}_{B_a} - 1} \right)$$

which does no longer depend on $\ell_a$. Dividing $\Psi_a$ by $\ell_a$ and $\Phi_a + \Psi_a + \Omega_a$ by $B_a$ yields the result. $\square$

Putting everything together, we have $\mathrm{PRD}/\mathrm{ALG} \leq \max_a \max \left\{ \tau_a, \max_{\ell_a \in \{1,\ldots,B_a\}} \Psi_a/\ell_a \right\}$ as $\tau_a$ does not depend on $\ell_a$. The reader can refer back to Figure 2 for an illustration of this upper bound. In the following lemma, we further analyze analytically $\max_{\ell_a \in \{1,\ldots,B_a\}} \Psi_a/\ell_a$ and compare it with $\tau_a$ to obtain the upper bound:

**Lemma 11.** *The consistency of Algorithm 1 is given by*

$$\mathrm{PRD}/\mathrm{ALG} \leq \left( 1 + \frac{1}{e^{\alpha}_B - 1} \max \left\{ \frac{1}{\alpha_B} \left( e^{\alpha}_B - \frac{e^{\alpha}_B - 1}{\alpha_B} \right), \ln (e^{\alpha}_B) \right\} \right).$$

*Proof.* Due to Lemma 9, it is sufficient to show

$$1 + \frac{1}{e^{\alpha}_B - 1} \max \left\{ \frac{1}{\alpha_B} \left( e^{\alpha}_B - \frac{e^{\alpha}_B - 1}{\alpha_B} \right), \ln (e^{\alpha}_B) \right\} \geq \begin{cases} \max \{ \tau_a, \Psi_a/\ell_a \} & \text{if } \ell_a > 0 \\ \tau_a & \text{otherwise.} \end{cases}$$

By Lemma 10, we know for the first term in the maximum that

$$\tau_a = 1 + \frac{1}{e^{\alpha}_{B_a} - 1} \frac{1}{\alpha_B} \left( e^{\alpha}_{B_a} - \frac{e^{\alpha}_{B_a} - 1}{\alpha_{B_a}} \right)$$

This term is maximized for $B_a = B$ since

$$1 + \frac{1}{e^{\alpha}_{B_a} - 1} \frac{1}{\alpha_B} \left( e^{\alpha}_{B_a} - \frac{e^{\alpha}_{B_a} - 1}{B_a \left( e^{\alpha/B_a}_{B_a} - 1 \right)} \right) = 1 + \frac{1}{\alpha_B} \left( \frac{e^{\alpha}_{B_a}}{e^{\alpha}_{B_a} - 1} - \frac{1}{\alpha_{B_a}} \right)$$

$$\leq 1 + \frac{1}{\alpha_B} \left( \frac{e^{\alpha}_B}{e^{\alpha}_B - 1} - \frac{1}{\alpha_B} \right) = 1 + \frac{1}{e^{\alpha}_B - 1} \underbrace{\frac{1}{\alpha_B} \left( e^{\alpha}_B - \frac{e^{\alpha}_B - 1}{\alpha_B} \right)}_{=:p(\alpha)}$$

due to Lemma 2. The lemma statement therefore follows immediately if $\ell_a = 0$. We may thus assume that $\ell_a > 0$ and use Lemma 10 to determine the second term in the maximum as

$$\frac{\Phi_a}{\ell_a} = 1 + \frac{1}{e^{\alpha}_{B_a} - 1} \left( \frac{1}{x} - 1 \right) \left( e^{\alpha x}_{B_a} - 1 \right)$$

where $x =: \ell_a/B_a$. The second term behaves similarly to the first as

$$\frac{\Phi_a}{\ell_a} = 1 + \frac{1}{e^{\alpha}_{B_a} - 1} \left( \frac{1}{x} - 1 \right) \left( e^{\alpha x}_{B_a} - 1 \right) \leq 1 + \frac{1}{e^{\alpha}_B - 1} \left( \frac{1}{x} - 1 \right) \left( e^{\alpha x}_B - 1 \right)$$

since $\left( e^{\alpha x}_{B_a} - 1 \right) / \left( e^{\alpha}_{B_a} - 1 \right) \leq \left( e^{\alpha x}_B - 1 \right) / \left( e^{\alpha}_B - 1 \right)$. We define $g(\alpha, x) := \left( \frac{1}{x} - 1 \right) \left( e^{\alpha x}_B - 1 \right)$ such that we can write

$$\max \left\{ \frac{\Phi_a + \Psi_a + \Omega_a}{B_a}, \frac{\Psi_a}{\ell_a} \right\} \leq 1 + \frac{1}{e^{\alpha}_B - 1} \max \{ p(\alpha), g(\alpha, x) \}.$$

We want to remove the dependency on $x$ in $g$ by maximizing $g$ over $x \in [0, 1]$ for a fixed $\alpha$. As $g(\alpha, x)$ is continuous, it suffices to evaluate $g$ in both endpoints and find the stationary points. We have

$$g(\alpha, 0) = \lim_{x \to 0} \left( \frac{1}{x} - 1 \right) \left( e_B^{\alpha x} - 1 \right) = \lim_{x \to 0} \frac{(1 - x)\left( e_B^{\alpha x} - 1 \right)}{x}$$

$$= \lim_{x \to 0} - \left( e_B^{\alpha x} - 1 \right) + (1 - x) \ln(e_B^{\alpha}) e_B^{\alpha x} = \ln(e_B^{\alpha})$$

by L'Hoptial. Further, $g(\alpha, 1) = 0$. Next, we find the stationary points $x^* \in [0, 1]$ as solutions to the equation

$$\frac{\partial}{\partial x} g(\alpha, x^*) = \ln(e_B^{\alpha}) \left( \frac{1}{x^*} - 1 \right) e_B^{\alpha x^*} - \frac{e_B^{\alpha x^*} - 1}{(x^*)^2} = 0$$

which is equivalent to

$$e_B^{\alpha x^*} - 1 = \ln(e_B^{\alpha})(x^*)^2 \left( \frac{1}{x^*} - 1 \right) e_B^{\alpha x^*}.$$

There is no closed form solution for $x^*$, but we can replace $e_B^{\alpha x} - 1$ in $g$ with the RHS of the above. This yields a new function

$$h(\alpha, y) = \left( \frac{1}{y} - 1 \right) \ln(e_B^{\alpha}) y^2 \left( \frac{1}{y} - 1 \right) e_B^{\alpha y} = \ln(e_B^{\alpha}) (1 - y)^2 e_B^{\alpha y}$$

with $h(\alpha, x^*) = g(\alpha, x^*)$. We can thus maximize $h$ over $y \in [0, 1]$ to obtain an upper bound on $g(x^*)$. Note that $h(\alpha, 0) = \ln(e_B^{\alpha}) = g(\alpha, 0)$ and $h(\alpha, 1) = 0 = g(\alpha, 1)$. To this end, let $y^*$ be such that

$$\frac{\partial}{\partial y^*} h(\alpha, y^*) = \ln(e_B^{\alpha})^2 (1 - y^*)^2 e_B^{\alpha y^*} - 2 \ln(e_B^{\alpha}) (1 - y^*) e_B^{\alpha y^*} = 0$$

which is equivalent to $\ln(e_B^{\alpha}) (1 - y^*) - 2 = 0$ or $y^* = 1 - \frac{2}{\ln(e_B^{\alpha})}$. We evaluate $h$ in $y^*$ and obtain

$$h(\alpha, y^*) = \alpha \ln(e_B) \left( \frac{2}{\alpha \ln(e_B)} \right)^2 e_B^{\alpha - \frac{2}{\ln(e_B)}} = \frac{4}{\alpha \ln(e_B) e^2} e_B^{\alpha}.$$

Note that $y^* \geq 0 \iff \alpha \geq 2 / \ln(e_B)$. Furthermore, $h^*(\alpha)$ always exceeds the endpoint $g(\alpha, 0)$: We calculate

$$h^*(\alpha) := h(\alpha, y^*) = \frac{4}{\ln(e_B^{\alpha}) e^2} e_B^{\alpha}$$

$$\geq \frac{4}{\ln(e_B^{\alpha}) e^2} e^2 \ln(e_B^{\alpha/2})^2$$

$$= \ln(e_B^{\alpha})$$

where the inequality is due to $e^z \geq ez$ for $z = \ln\left( e_B^{\alpha/2} \right) \geq 0$. Therefore, for all $x \in [0, 1]$,

$$g(\alpha, x) \leq \begin{cases} \ln(e_B^{\alpha}) & \text{if } \alpha \leq \frac{2}{\ln(e_B)} \\ h^*(\alpha) & \text{otherwise.} \end{cases}$$

We consider both intervals separately. Let us first consider the the case when $\alpha \in \left[ 0, \frac{2}{\ln(e_B)} \right]$. If $B < \infty$, there could be multiple intersection points between $p(\alpha)$ and $\alpha \ln(e_B)$. However, the situation is easier if $B \to \infty$ as the intersection points given by

$$p(\alpha) = \frac{1}{\alpha} \left( e^{\alpha} - \frac{e^{\alpha} - 1}{\alpha} \right) = \alpha \iff \alpha e^{\alpha} - e^{\alpha} + 1 = \alpha^3$$

are at $\alpha = 1$ and $\alpha^* \approx 1.79$, whereas $\alpha \ln(e_B)$ dominates $p(\alpha)$ between $1$ and $\alpha^*$.

It remains to consider the case $\alpha \geq \frac{2}{\ln(e_{B_a})}$. Again, there can be many intersection points of $p(\alpha)$ with $\alpha \ln(e_{B_a})$ and $h^*(\alpha)$. However, if $B \to \infty$, then $p(\alpha)$ already dominates $h^*(\alpha)$ for $\alpha > 2$ which we can see as follows. First,

$$h^*(\alpha) = \frac{4}{\alpha} e^{\alpha - 2} \leq \frac{1}{\alpha} \left( e^{\alpha} - \frac{e^{\alpha} - 1}{\alpha} \right) = p(\alpha)$$

$$\Longleftrightarrow 4e^{-2} \leq 1 - \frac{1 - e^{-\alpha}}{\alpha}.$$

We can see that $\frac{1-e^{-\alpha}}{\alpha}$ is decreasing in $\alpha$ as

$$\frac{\partial}{\partial \alpha} \frac{1 - e^{-\alpha}}{\alpha} = \frac{e^{-\alpha} (\alpha - e^{\alpha} + 1)}{\alpha^2} \leq 0$$

which holds as $1 + \alpha \leq e^{\alpha}$. Finally, we check that $h^*(2) = 2 \leq 2.10 \approx p(2)$. $\qquad \square$

# B  Generalized Assignment Problem

The generalized assignment problem (GAP) is a generalization of Display Ads where impressions $t$ can take up any size $u_{at}$ in the budget constraint of advertiser $a$. This formulation encompasses both Display Ads and Ad Words, and we empirically compare it to the Ad Words algorithm with predictions due to Mahdian et al. (2007) in Section C.3. For simplicity of presentation, we assume that budgets are all 1 and instead, $u_{at} \to 0$. However, as before it is possible to adapt the algorithm to work with large sizes $u_{at}$. We state the LP below.



GAP Primal         GAP Dual

$$\max \sum_{a,t} w_{at} x_{at} \qquad\qquad \min \sum_a \beta_a + \sum_t z_t$$

$$\forall a : \sum_t u_{at} x_{at} \leq 1 \qquad \forall a, t : z_t \geq w_{at} - u_{at} \beta_a$$

$$\forall t : \sum_a x_{at} \leq 1$$



Algorithm 2 is a generalization of Algorithm 1 to GAP. An immediate difference is that the discounted gain $w_{at} - u_{at}\beta_a$ respects the impression size $u_{at}$ in accordance with the changed dual constraint. We still follow the predicted advertiser if its discounted gain still is a sufficiently high fraction of the maximum discounted gain. However, we might now have to remove multiple impressions with least value-size ratio to accommodate the new impression. The update for $\beta_a$ also differs and is based on value-size ratios of impressions allocated to $a$: For a fixed advertiser $a$ let $U_a = \sum_{t \in \mathbf{X}_a} u_{at}$, be the total size of all impressions ever allocated to $a$. For any $x \in (0, U_a]$ define $\frac{w_x}{u_x}$ as the minimal ratio such that

$$\sum_{t \in \mathbf{X}_a : \frac{w_{at}}{u_{at}} \leq \frac{w_x}{u_x}} u_{at} > x. \tag{12}$$

Then, we can naturally define $\beta_a$ as the exponential average over ratios $\frac{w_x}{u_x}$. As before, we also assume that there exists a dummy advertiser that only receives impressions of zero value-size ratio and that all advertisers are initially filled up with impressions of zero value.

## B.1  Robustness

**Theorem 12.** *Algorithm 1 has a robustness of*

$$\frac{\mathrm{ALG}}{\mathrm{OPT}} \geq \frac{e^{\alpha} - 1}{\alpha e^{\alpha}}$$

*Proof.* Assume we assign impression $t$ to advertiser $a$ while disposing of some impressions to make space. We will bound the dual increase as a multiple of the primal increase. We now assume that after allocating $t$ to $a$, it becomes the impression with highest value-size ratio (a general proof follows analogously to the proof of robustness for Display Ads in Section A.1). The primal increase is simply

$$\Delta P = \int_{U_a - u_{at}}^{U_a} \frac{w_x}{u_x} dx - \int_{U_a - 1 - u_{at}}^{U_a - 1} \frac{w_x}{u_x} dx = w_{at} - \int_{U_a - 1 - u_{at}}^{U_a - 1} \frac{w_x}{u_x} dx.$$

---

**Algorithm 2** Exponential Averaging with Predictions for GAP

---

1: **Input:** Robustness-consistency trade-off parameter $\alpha \in [1, \infty)$
2: For each advertiser $a$, initialize $\beta_a \leftarrow 0$ and fill up $a$ with zero-value impressions
3: **for all** arriving impressions $t$ **do**
4:     $a_{(\mathrm{PRD})} \leftarrow \mathrm{PRD}(t)$
5:     $a_{(\mathrm{EXP})} \leftarrow \arg\max_a \{w_{at} - u_{at}\beta_a\}$
6:     **if** $\alpha \left(w_{a_{(\mathrm{PRD})},t} - u_{a_{(\mathrm{PRD})},t}\beta_{a_{(\mathrm{PRD})}}\right) \geq w_{a_{(\mathrm{EXP})},t} - u_{a_{(\mathrm{EXP})},t}\beta_{a_{(\mathrm{EXP})}}$ **then**
7:       $a \leftarrow a_{(\mathrm{PRD})}$
8:     **else**
9:       $a \leftarrow a_{(\mathrm{EXP})}$
10:    **end if**
11:    Dispose of impressions with least value-size ratio currently allocated to $a$ until there is $u_{at}$ of free space and allocate $t$ to $a$
12:    Let $\frac{w_x}{u_x}$ as in (12) and update $\beta_a \leftarrow \dfrac{\alpha}{e^\alpha - 1}\displaystyle\int_{U_a-1}^{U_a} \frac{w_x}{u_x} e^{\alpha(U_a - x)}dx$
13: **end for**

---

At the same time,

$$\beta_a^{(t)} = \frac{\alpha}{e^\alpha - 1}\int_{U_a-1}^{U_a} \frac{w_x}{u_x} e^{\alpha(U_a-x)}dx$$

$$= \frac{\alpha}{e^\alpha - 1}\left(\int_{U_a-1-u_{at}}^{U_a-u_{at}} \frac{w_x}{u_x}e^{\alpha(U_a-x)}dx + \int_{U_a-u_{at}}^{U_a} \frac{w_x}{u_x}e^{\alpha(U_a-x)}dx - \int_{U_a-1-u_{at}}^{U_a-1} \frac{w_x}{u_x}e^{\alpha(U_a-x)}dx\right)$$

$$= \frac{\alpha}{e^\alpha - 1}\left(e^{\alpha u_{at}}\int_{U_a-1-u_{at}}^{U_a-u_{at}} \frac{w_x}{u_x}e^{\alpha(U_a-x-u_{at})}dx + w_{at} - \int_{U_a-1-u_{at}}^{U_a-1} \frac{w_x}{u_x}e^{\alpha(U_a-x)}dx\right)$$

$$= e^{\alpha u_{at}}\beta_a^{(t-1)} + \frac{\alpha}{e^\alpha - 1}\left(w_{at} - \int_{U_a-1-u_{at}}^{U_a-1} \frac{w_x}{u_x}e^{\alpha(U_a-x)}dx\right)$$

We set $z_t = \alpha\left(w_{at} - u_{at}\beta_a^{(t-1)}\right)$ and obtain, since $e^{\alpha u_{at}} - 1 = \alpha u_{at}$ due to $u_{at} \to 0$,

$$\Delta D = \beta_a^{(t)} - \beta_a^{(t-1)} + z_t$$

$$= (e^{\alpha u_{at}} - 1)\beta_a^{(t-1)} + \frac{\alpha}{e^\alpha - 1}\left(w_{at} - \int_{U_a-1-u_{at}}^{U_a-1} \frac{w_x}{u_x}e^{\alpha(U_a-x)}dx\right) + \alpha\left(w_{at} - u_{at}\beta_a^{(t-1)}\right)$$

$$= \alpha u_{at}\beta_a^{(t-1)} + \frac{\alpha}{e^\alpha - 1}\left(w_{at} - \int_{U_a-1-u_{at}}^{U_a-1} \frac{w_x}{u_x}e^{\alpha(U_a-x)}dx\right) + \alpha\left(w_{at} - u_{at}\beta_a^{(t-1)}\right)$$

$$= \frac{\alpha e^\alpha}{e^\alpha - 1}w_{at} - \frac{\alpha e^\alpha}{e^\alpha - 1}\int_{U_a-1-u_{at}}^{U_a-1} \frac{w_x}{u_x}dx$$

$$= \frac{\alpha e^\alpha}{e^\alpha - 1}\Delta P.$$

$\square$

## B.2   Consistency

**Theorem 13.** *Algorithm 1 has a consistency of*

$$\frac{\mathrm{ALG}}{\mathrm{PRD}} \geq \left(1 + \frac{1}{e^\alpha - 1}\max\left\{\frac{1}{\alpha}\left(e^\alpha - \frac{e^\alpha - 1}{\alpha}\right), \alpha\right\}\right)^{-1}.$$

As before, we split the impressions $t$ based on whether the algorithm followed the prediction or not. If the algorithm ignores the prediction, we can use that $\alpha\left(w_{a_{(\mathrm{PRD})},t} - u_{a_{(\mathrm{PRD})},t}\beta_{a_{(\mathrm{PRD})}}\right) \leq$

$w_{a_{(\mathrm{EXP})},t} - u_{a_{(\mathrm{EXP})},t}\beta_{a_{(\mathrm{EXP})}}$ due to Line 6 in Algorithm 2. With a similar calculation, we obtain

$$\mathrm{PRD} = \sum_a \mathrm{PRD}_a$$

$$= \sum_a \left( \sum_{t \in \mathbf{P}_a \cap \mathbf{X}_a} w_{at} + \frac{1}{\alpha} \sum_{t \in \mathbf{X}_a \setminus \mathbf{P}_a} w_{at} - \frac{1}{\alpha} \sum_{t \in \mathbf{X}_a \setminus \mathbf{P}_a} u_{at}\beta_a^{(t-1)} + \sum_{t \in \mathbf{P}_a \setminus \mathbf{X}_a} u_{at}\beta_a^{(t-1)} \right).$$

Once again, we fix an advertiser $a$. Let $\rho_a \coloneqq \sum_{t \in \mathbf{P}_a \cap \mathbf{X}_a} u_{at}$ so that we can bound

$$\sum_{t \in \mathbf{P}_a \setminus \mathbf{X}_a} u_{at}\beta_a^{(t-1)} \leq (1 - \rho_a)\,\beta_a^{(T)}.$$

We still have to argue that the worst-case is when all impressions are ordered such that their value-size ratios are non-decreasing and the impressions in $\mathbf{P}_a \cap \mathbf{X}_a$ are the ones with maximum value-size ratio among $\mathbf{X}_a$. The latter is obvious as it can only increase the value of PRD, so it remains to show that the non-decreasing value-size ordering minimizes the third sum in $\mathrm{PRD}_a$ (the first two sums are invariant under reordering). To this end, note that the value of $\beta_a$ for GAP is the limit of $\beta_a$ for Display Ads in the following sense: For positive $\epsilon \to 0$, we can split each GAP-impression $t \in \mathbf{X}_a$ into $\frac{u_t}{\epsilon}$ identical Display Ads-impressions with value $\frac{w_t}{u_t}$, while assuming a budget of $1/\epsilon$. Then, the GAP $\beta_a$ and Display Ads $\beta_a$ are identical. As we know from Display Ads, the worst case is achieved when the Display Ads-impressions with value $\frac{w_t}{u_t}$ are in non-decreasing order. In this ordering, consecutive Display-Ads impressions with identical value $\frac{w_t}{u_t}$ still correspond to the same GAP-impression $t$, so we also know that this ordering is the worst-case for GAP. We may therefore assume that the impressions are ordered such that their value-size ratios are non-decreasing. As such, we obtain

$$\beta_a^{(t)} = \frac{\alpha}{e^\alpha - 1} \int_{U_a^{(t)}-1}^{U_a^{(t)}} \frac{w_x}{u_x} e^{\alpha(U_a^{(t)}-x)} dx = \int_{y-1}^{y} \frac{w_x}{u_x} e^{\alpha(y-x)} dx =: \beta_a^{(y)}$$

where $y = U_a^{(t)}$. Combined with the fact that $\mathbf{P}_a \cap \mathbf{X}_a$ are last impressions in $\mathbf{X}_a$, we can now write

$$\sum_{t \in \mathbf{P}_a \cap \mathbf{X}_a} w_{at} + \frac{1}{\alpha} \sum_{t \in \mathbf{X}_a \setminus \mathbf{P}_a} w_{at} - \frac{1}{\alpha} \sum_{t \in \mathbf{X}_a \setminus \mathbf{P}_a} u_{at}\beta_a^{(t-1)}$$

$$= \int_{U_a-\rho}^{U_a} \frac{w_x}{u_x} dx + \frac{1}{\alpha} \int_0^{U_a-\rho_a} \frac{w_x}{u_x} dx - \frac{1}{\alpha} \int_0^{U_a-\rho_a} \beta_a^{(x)} dx$$

This helps us to compute $\beta_a^{(x)}$ in $\mathrm{PRD}_a$ and rewrite the whole term as a linear combination of value-size ratios.

**Lemma 14.** *We have*

$$\mathrm{PRD}_a \leq \int_{U_a-1}^{U_a-\rho_a} \frac{w_x}{u_x} \phi_x dx + \int_{U_a-\rho_a}^{U_a} \frac{w_x}{u_x} \psi_x dx + \frac{w_{U_a-1}}{u_{U_a-1}} \Omega_a$$

*where*

$$\phi_x \coloneqq (1 - \rho_a) \frac{\alpha}{e^\alpha - 1} e^{\alpha(U_a-x)} + \frac{1}{\alpha} \frac{e^\alpha - e^{\alpha(U_a-\rho_a-x)}}{e^\alpha - 1}$$

$$\psi_x \coloneqq 1 + (1 - \rho_a) \frac{\alpha}{e^\alpha - 1} e^{\alpha(U_a-x)}$$

$$\Omega_a \coloneqq \frac{1}{\alpha} \frac{1}{e^\alpha - 1} \left( \rho_a e^\alpha - \frac{1}{\alpha} \left( e^\alpha - e^{\alpha(1-\rho_a)} \right) \right)$$

*Proof.* We rewrite the third sum in $\mathrm{PRD}_a$ to

$$\int_0^{U_a-\rho_a} \beta_a^{(x)} dx$$

$$= \frac{\alpha}{e^{\alpha} - 1} \int_0^{U_a - \rho_a} \int_{x-1}^{x} \frac{w_y}{u_y} e^{\alpha(x-y)} dy dx$$

$$= \frac{\alpha}{e^{\alpha} - 1} \int_0^{U_a - \rho_a} \frac{w_y}{u_y} \int_0^{\min\{1, U_a - \rho_a - y\}} e^{\alpha x} dx dy$$

$$= \frac{\alpha}{e^{\alpha} - 1} \int_0^{U_a - 1 - \rho_a} \frac{w_y}{u_y} \int_0^{1} e^{\alpha x} dx dy + \frac{\alpha}{e^{\alpha} - 1} \int_{U_a - 1 - \rho_a}^{U_a - \rho_a} \frac{w_y}{u_y} \int_0^{U_a - \rho_a - y} e^{\alpha x} dx dy$$

$$= \int_0^{U_a - 1 - \rho_a} \frac{w_y}{u_y} dy + \frac{1}{e^{\alpha} - 1} \int_{U_a - 1 - \rho_a}^{U_a - \rho_a} \frac{w_y}{u_y} \left( e^{\alpha(U_a - \rho_a - y)} - 1 \right) dy$$

where for the last equality, we simply evaluated the integral. Using this in place of the second sum in $\mathrm{PRD}_a$ cancels out most of the terms of the second sum:

$$\int_0^{U_a - \rho_a} \frac{w_x}{u_x} dx - \int_0^{U_a - \rho_a} \beta_a^{(x)} dx$$

$$= \int_0^{U_a - \rho_a} \frac{w_x}{u_x} dx - \int_0^{U_a - 1 - \rho_a} \frac{w_y}{u_y} dy - \frac{1}{e^{\alpha} - 1} \int_{U_a - 1 - \rho_a}^{U_a - \rho_a} \frac{w_y}{u_y} \left( e^{\alpha(U_a - \rho_a - y)} - 1 \right) dy$$

$$= \int_{U_a - 1 - \rho_a}^{U_a - \rho_a} \frac{w_y}{u_y} \left( 1 - \frac{e^{\alpha(U_a - \rho_a - y)} - 1}{e^{\alpha} - 1} \right) dy$$

$$= \int_{U_a - 1 - \rho_a}^{U_a - \rho_a} \frac{w_y}{u_y} \frac{e^{\alpha} - e^{\alpha(U_a - \rho_a - y)}}{e^{\alpha} - 1} dy$$

$$= \int_{U_a - 1}^{U_a - \rho_a} \frac{w_y}{u_y} \frac{e^{\alpha} - e^{\alpha(U_a - \rho_a - y)}}{e^{\alpha} - 1} dy + \int_{U_a - 1 - \rho_a}^{U_a - 1} \frac{w_y}{u_y} \frac{e^{\alpha} - e^{\alpha(U_a - \rho_a - y)}}{e^{\alpha} - 1} dy. \tag{13}$$

We upper bound the third sum

$$\frac{1}{\alpha} \int_{U_a - 1 - \rho_a}^{U_a - 1} \frac{w_y}{u_y} \frac{e^{\alpha} - e^{\alpha(U_a - \rho_a - y)}}{e^{\alpha} - 1} dy \leq \frac{w_{U_a - 1}}{u_{U_a - 1}} \frac{1}{\alpha} \int_{U_a - 1 - \rho_a}^{U_a - 1} \frac{e^{\alpha} - e^{\alpha(U_a - \rho_a - y)}}{e^{\alpha} - 1} dy$$

$$= \frac{w_{U_a - 1}}{u_{U_a - 1}} \frac{1}{\alpha} \frac{1}{e^{\alpha} - 1} \left( \rho_a e^{\alpha} - \int_{1 - \rho_a}^{1} e^{\alpha y} dy \right)$$

$$= \frac{w_{U_a - 1}}{u_{U_a - 1}} \underbrace{\frac{1}{\alpha} \frac{1}{e^{\alpha} - 1} \left( \rho_a e^{\alpha} - \frac{1}{\alpha} \left( e^{\alpha} - e^{\alpha(1 - \rho_a)} \right) \right)}_{= \Omega_a} \tag{14}$$

Furthermore,

$$(1 - \rho_a) \beta_a^{(U_a)} = (1 - \rho_a) \frac{\alpha}{e^{\alpha} - 1} \int_{U_a - 1}^{U_a} \frac{w_x}{u_x} e^{\alpha(U_a - x)} dx \tag{15}$$

Combining (13), (14), and (15) and grouping terms yields

$$\int_{U_a - \rho_a}^{U_a} \frac{w_x}{u_x} dx + \frac{1}{\alpha} \int_{U_a - 1}^{U_a - \rho_a} \frac{w_y}{u_y} \frac{e^{\alpha} - e^{\alpha(U_a - \rho_a - y)}}{e^{\alpha} - 1} dy + \frac{w_{U_a - 1}}{u_{U_a - 1}} \Omega_a$$

$$+ (1 - \rho_a) \frac{\alpha}{e^{\alpha} - 1} \int_{U_a - 1}^{U_a} \frac{w_x}{u_x} e^{\alpha(U_a - x)} dx$$

$$= \int_{U_a - 1}^{U_a - \rho_a} \frac{w_x}{u_x} \underbrace{\left( (1 - \rho_a) \frac{\alpha}{e^{\alpha} - 1} e^{\alpha(U_a - x)} + \frac{1}{\alpha} \frac{e^{\alpha} - e^{\alpha(U_a - \rho_a - x)}}{e^{\alpha} - 1} \right)}_{= \phi_x} dx$$

$$+ \int_{U_a - \rho_a}^{U_a} \frac{w_x}{u_x} \underbrace{\left( 1 + (1 - \rho_a) \frac{\alpha}{e^{\alpha} - 1} e^{\alpha(U_a - x)} \right)}_{= \psi_x} dx + \frac{w_{U_a - 1}}{u_{U_a - 1}} \Omega_a.$$

$\square$

Analogously to Display Ads, we define

$$\Phi_a := \int_{U_a-1}^{U_a-\rho_a} \phi_x dx \qquad \text{and} \qquad \Psi_a := \int_{U_a-\rho}^{U_a} \psi_x dx$$

and the total coefficient $\tau_a := \Phi_a + \Psi_a + \Omega_a$ which by a calculation similar to Lemma 10 can be shown to be

$$\tau_a = 1 + \frac{1}{e^\alpha - 1}\frac{1}{\alpha}\left(e^\alpha - \frac{e^\alpha - 1}{\alpha}\right).$$

**Lemma 15.** *We have*

$$\text{PRD} \le \max\left\{\tau_a, \frac{\Psi_a}{\rho_a}\right\} \text{ALG}$$

*if $\rho_a > 0$ and otherwise,*

$$\text{PRD} \le \tau_a \text{ALG}.$$

*Proof.* Again, let

$$\bar{w}_\Phi := \frac{1}{1-\rho_a}\int_{U_a-1}^{U_a-\rho_a} \frac{w_x}{u_x}dx$$

$$\bar{w}_\Psi := \frac{1}{\rho_a}\int_{U_a-\rho_a}^{U_a} \frac{w_x}{u_x}dx$$

be the average coefficients on the intervals $[U_a - 1, U_a - \rho_a]$ and $[U_a - \rho_a, U_a]$, respectively. The latter coefficients are still decreasing as

$$\psi_x = 1 + \underbrace{(1-\rho_a)\frac{\alpha}{e^\alpha - 1}}_{\ge 0} e^{\alpha(U_a - x)}$$

so we can bound the linear combination

$$\int_{U_a-\rho_a}^{U_a} \frac{w_x}{u_x}\psi_x dx \le \bar{w}_\Psi \int_{U_a-\rho}^{U_a} \psi_x dx = \bar{w}_\Psi \Psi_a.$$

However, $\phi_x$ is not always decreasing which can be seen by rearranging

$$\phi_x = (1-\rho_a)\frac{\alpha}{e^\alpha - 1}e^{\alpha(U_a-x)} + \frac{1}{\alpha}\frac{e^\alpha - e^{\alpha(U_a-\rho_a-x)}}{e^\alpha - 1}$$

$$= \frac{1}{e^\alpha - 1}\left((1-\rho_a)\alpha - \frac{1}{\alpha}e^{-\alpha\rho_a}\right)e^{\alpha(U_a-x)} + \frac{1}{\alpha}\frac{1}{e^\alpha - 1}e^\alpha$$

We observe that $\phi_x$ is decreasing if $(1-\rho_a)\alpha$ is at least $\frac{1}{\alpha}e^{-\alpha\rho_a}$, and we analyze two cases based on the relationship of both terms:

- $(1-\rho_a)\alpha \ge \frac{1}{\alpha}e^{-\alpha\rho_a}$: We have $\int_{U_a-1}^{U_a-\rho_a} \frac{w_x}{u_x}\phi_x dx \le \bar{w}_\Phi \Phi_a$ and thus

$$\int_{U_a-1}^{U_a-\rho_a} \frac{w_x}{u_x}\phi_x dx + \int_{U_a-\rho}^{U_a} \frac{w_x}{u_x}\psi_x dx + \frac{w_{U_a-1}}{u_{U_a-1}}\Omega_a$$

$$\le \bar{w}_\Phi \Phi_a + \bar{w}_\Psi \Psi_a + w_{a,I_a-B_a}\Omega_a$$

$$\le \bar{w}_\Phi\left(\Phi_a + \Omega_a\right) + \bar{w}_\Psi \Psi_a$$

$$= \bar{w}_\Phi\left(1-\rho_a\right)\tau_a + \bar{w}_\Phi\left(\Phi_a + \Omega_a - (1-\rho_a)\tau_a\right) + \bar{w}_\Psi \Psi_a$$

$$= \int_{U_a-1}^{U_a-\rho_a} \frac{w_x}{u_s}\tau_a dx + \bar{w}_\Phi\left(\Phi_a + \Omega_a - (1-\rho_a)\tau_a\right) + \bar{w}_\Psi \Psi_a \qquad (16)$$

- $(1-\rho_a)\alpha \le \frac{1}{\alpha}e^{-\alpha\rho_a}$: We can still show that $\phi_x \le \tau_a$ as

$$\phi_x = (1 - \rho_a) \frac{\alpha}{e^\alpha - 1} e^{\alpha(U_a - x)} + \frac{1}{\alpha} \frac{e^\alpha - e^{\alpha(U_a - \rho_a - x)}}{e^\alpha - 1}$$

$$\leq 1 + \frac{1}{e^\alpha - 1} \frac{1}{\alpha} \left( e^\alpha - \frac{e^\alpha - 1}{\alpha} \right) = \tau_a$$

$$\Longleftrightarrow \underbrace{\left( (1 - \rho_a) \alpha - \frac{1}{\alpha} e^{-\alpha \rho_a} \right)}_{\leq 0} \underbrace{e^{\alpha(U_a - x)}}_{\geq 0} \leq e^\alpha - 1 - \frac{1}{\alpha} \frac{e^\alpha - 1}{\alpha} = \underbrace{\left( 1 - \frac{1}{\alpha^2} \right)}_{\geq 0} \underbrace{(e^\alpha - 1)}_{\geq 0}.$$

Therefore,

$$\int_{U_a-1}^{U_a-\rho_a} \frac{w_x}{u_x} \phi_x dx + \int_{U_a-\rho}^{U_a} \frac{w_x}{u_x} \psi_x dx + \frac{w_{U_a-1}}{u_{U_a-1}} \Omega_a$$

$$\leq \int_{U_a-1}^{U_a-\rho_a} \frac{w_x}{u_x} \phi_x dx + \bar{w}_\Psi \Psi_a + \frac{w_{U_a-1}}{u_{U_a-1}} \Omega_a$$

$$= \int_{U_a-1}^{U_a-\rho_a} \frac{w_x}{u_x} \tau_a dx - \int_{U_a-1}^{U_a-\rho_a} \frac{w_x}{u_x} (\tau_a - \phi_x) dx + \bar{w}_\Psi \Psi_a + \frac{w_{U_a-1}}{u_{U_a-1}} \Omega_a$$

$$\leq \int_{U_a-1}^{U_a-\rho_a} \frac{w_x}{u_x} \tau_a dx - \int_{U_a-1}^{U_a-\rho_a} \frac{w_{U_a-1}}{u_{U_a-1}} (\tau_a - \phi_x) dx + \bar{w}_\Psi \Psi_a + \frac{w_{U_a-1}}{u_{U_a-1}} \Omega_a$$

$$= \int_{U_a-1}^{U_a-\rho_a} \frac{w_x}{u_x} \tau_a dx + \frac{w_{U_a-1}}{u_{U_a-1}} (\Phi_a + \Omega_a - (1 - \rho_a) \tau_a) + \bar{w}_\Psi \Psi_a \tag{17}$$

In both cases (16) and (17), we have shown that

$$\int_{U_a-1}^{U_a-\rho_a} \frac{w_x}{u_x} \phi_x dx + \int_{U_a-\rho}^{U_a} \frac{w_x}{u_x} \psi_x dx + \frac{w_{U_a-1}}{u_{U_a-1}} \Omega_a$$

$$\leq \int_{U_a-1}^{U_a-\rho_a} \frac{w_x}{u_x} \tau_a dx + v (\Phi_a + \Omega_a - (1 - \rho_a) \tau_a) + \bar{w}_\Psi \Psi_a$$

for a $v \leq \bar{w}_\Phi$.

$$\int_{U_a-1}^{U_a-\rho_a} \frac{w_x}{u_x} \tau_a dx + v (\Phi_a + \Omega_a - (1 - \rho_a) \tau_a) + \bar{w}_\Psi \Psi_a$$

$$\leq \int_{U_a-1}^{U_a-\rho_a} \frac{w_x}{u_x} \tau_a dx + \bar{w}_\Psi \max \{ \Phi_a + \Omega_a - (1 - \rho_a) \tau_a, 0 \} + \bar{w}_\Psi \Psi_a$$

$$= \int_{U_a-1}^{U_a-\rho_a} \frac{w_x}{u_x} \tau_a dx + \bar{w}_\Psi \max \{ \Phi_a + \Psi_a + \Omega_a - (1 - \rho_a) \tau_a, \Psi_a \}$$

$$= \int_{U_a-1}^{U_a-\rho_a} \frac{w_x}{u_x} \tau_a dx + \bar{w}_\Psi \max \{ \rho_a \tau_a, \Psi_a \}$$

$$\leq \tau_a \int_{U_a-1}^{U_a-\rho_a} \frac{w_x}{u_x} dx + \max \left\{ \tau_a, \frac{\Psi_a}{\rho_a} \right\} \int_{U_a-\rho_a}^{U_a} \frac{w_x}{u_x} dx$$

$$\leq \max \left\{ \tau_a, \frac{\Psi_a}{\rho_a} \right\} \int_{U_a-1}^{U_a} \frac{w_x}{u_x} dx$$

or $\leq \tau_a \int_{U_a-1}^{U_a} \frac{w_x}{u_x} dx$ if $\rho_a = 0$ $\qquad\qquad\qquad\qquad\qquad\qquad\qquad\qquad$ $\square$

Note that for the bound of Lemma 11, we did not require that $\ell_a$ is integral. We can thus apply Lemma 11 to bound $\max \left\{ \tau_a, \frac{\Psi_a}{\rho_a} \right\}$ and obtain the same result, which proves Theorem 13.

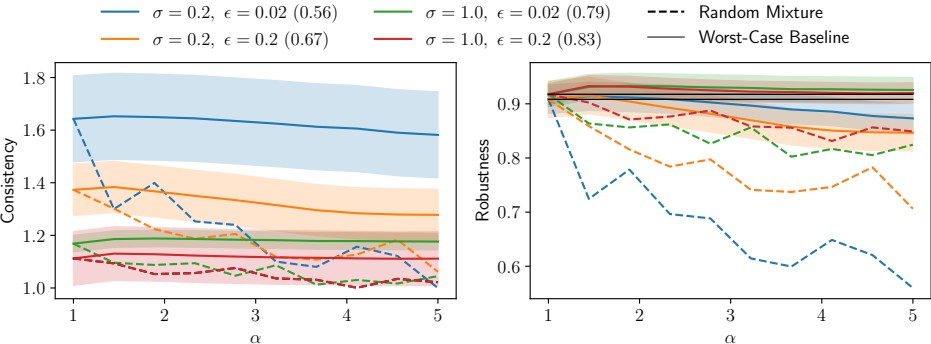

Figure 6: Experimental results for varying values of $\alpha$ on synthetic data with 12 advertisers and 2000 impressions of 10 types, where we report the same quantities as in Figure 3. We use Dual Base predictions for different $\sigma$ and $\epsilon$. Note that there are two black lines indicating the performance of the worst-case algorithm without predictions, corresponding to the datasets with differing $\sigma$.

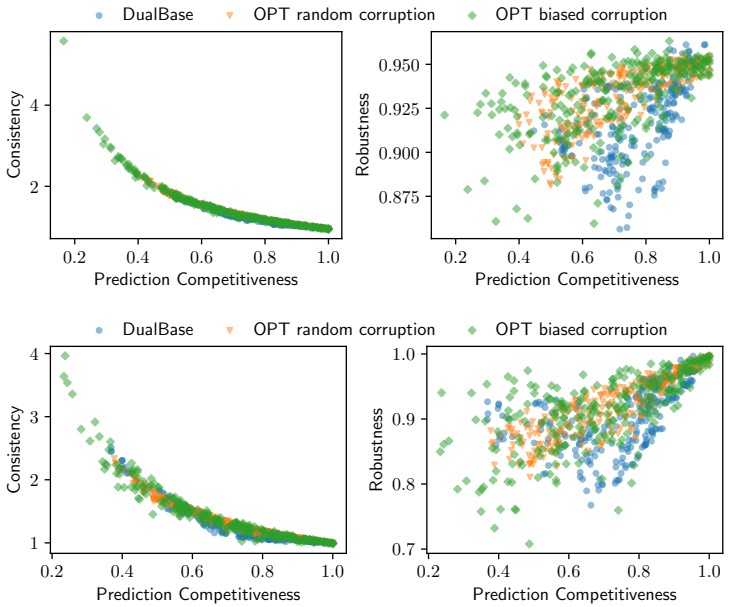

Figure 7: Performance for varying prediction quality with the data from Figure 6 (top) for $\alpha = 2$ (top) and $\alpha = 5$ (bottom).

## C  Further Experimental Results

### C.1  Real-World Data

**Description of the Yahoo Dataset:** The original dataset contains impression allocations to 16268 advertisers throughout 123 days, each tagged with the advertiser that bought the impression and a set of keyphrases that categorize the user for whom the impression is displayed. Lavastida et al. (2021) then consider the 20 most common keyphrases and create an impression type for each non-empty subset thereof. Whenever an advertiser buys an impression with a certain set of keyphrases, we assume that all impression types that correspond to a superset of these keyphrases are relevant for this advertiser, and that it derives some constant value (say, 1) from this allocation. At the same time, the number of impressions we create from each impression type (i.e. the supply) is the number of impression allocations in the original dataset that show that the impression type is relevant for an advertiser. As such, we obtain around 2 million impressions. Lavastida et al. (2021) try multiple

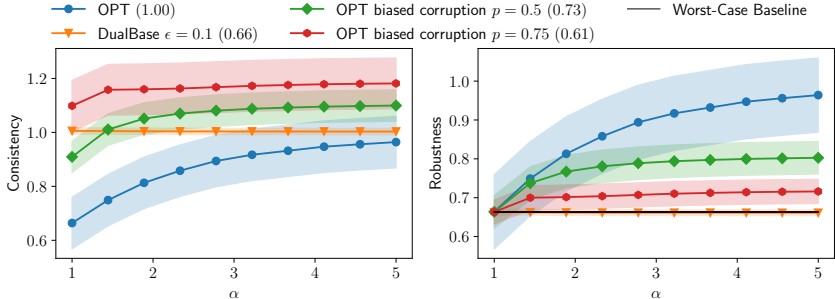

Figure 8: Performance on a worst-case instance with different predictors.

impression orders and budgets for the advertisers, but due to space constraints we restrict ourselves to display all impressions of a type at once, in supply-ascending order. We determine advertisers' budgets by allocating each impression to one of the advertisers with non-zero valuation uniformly at random and taking the number of allocated impressions at the end to be the advertiser's budget.

## C.2 Synthetic Data

**Results:** Figure 5 shows consistency and robustness of our algorithm on synthetic data on $T = 2000$ impressions of 10 types and $k = 12$ advertisers, for a variation of predictions. The plot shows the performance for predictions from the optimum solutions (with varying corruption) and the dual base prediction. Our algorithm converges to almost perfect consistency and robustness for $\alpha = 10$, given the optimum solution. At the same time, we observe that the algorithm is robust against both random and biased corruption, as the robustness does not drop to the prediction's low competitiveness of around $0.7$. Furthermore, the algorithm performs well in combination with the dual base prediction for $\epsilon = 0.1$ even though the first 200 impressions are clearly not representative of all synthetically generated impressions.

To investigate the our algorithm in conjunction with an easily available prediction, we also analyze the behavior of the dual base algorithm for different values of $\sigma$ and $\epsilon$ in Figure 6. The performance of our algorithm under dual base predictions clearly improves for increasing values of $\sigma$ as impressions become more evenly distributed across the day. Generally, sampling more impressions helps but dual base predictions may also lead to a drop in robustness, and more samples can even lead to a more adversarial prediction, as we explore further below. Yet, the robustness does still stays above the prediction's competitiveness in these cases.

Figure 7 shows consistency and robustness for different predictions with varying competitiveness on $\alpha \in \{2, 5\}$. We achieve this by varying the fraction $\epsilon \in [0, 1]$ of samples for the dual base algorithm and the corruption rate $p \in [0, 1]$ for random and biased corruptions. For $\alpha = 2$, the consistency exceeds 1 if the prediction is not very good (competitiveness below $0.9$). The algorithm is not heavily influenced by a bad prediction since $\alpha = 2$ is low, so the total obtained value remains relatively constant. For $\alpha = 5$, the algorithm might however follow the bad choices of the prediction, so the competitiveness varies more. As expected, the average robustness decreases for increasing $\alpha$, but the dual base prediction starts out with a much lower robustness than the corrupted predictions. The reason for that is that both the dual base algorithm and exponential averaging make their decisions based on the discounted gain. Our algorithm might therefore easily disregard a corrupted prediction as its discounted gain is low (or even negative), but the dual base prediction looks like a sensible choice. The dual base algorithm therefore manages to fool the algorithm for low $\alpha$, while a biased corruption leads to the worst corruption for larger values of $\alpha$.

**Hard Instances:** We consider the worst-case instance for the Display Ads problem described in Mehta et al. (2007). For $k$ advertisers, we create impressions of types $r \in \{1, \ldots, k\}$. An impression $t$ of type $r$ has zero value for the first $r - 1$ advertisers $w_{1,t} = \cdots = w_{r-1,t} = 0$ and value 1 for the following advertisers $w_{r,t} = \cdots = w_{k,t} = 1$. We first show all impressions of type 1, then all impressions of type 2, and so forth. The instance is difficult as the algorithm—not knowing about future impressions—has to allocate impressions of a type equally among advertisers that can

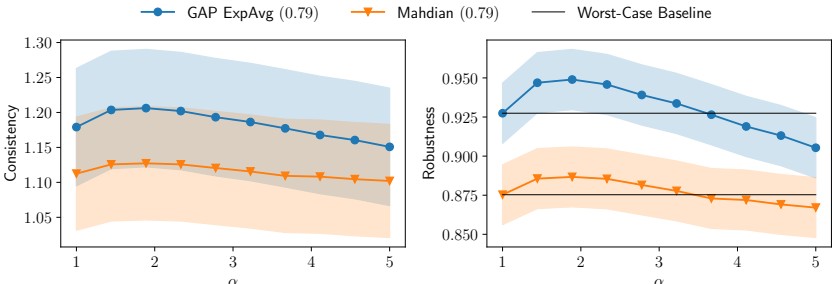

Figure 9: Performance on synthetic Ad Words instances, compared to the algorithm of Mehta et al. (2007). The black lines show the robustness of two worst-case algorithms without predictions: The algorithm due to Feldman et al. (2009a) which is the basis for our algorithm, and the algorithm of Mehta et al. (2007), which serves as a basis for the algorithm of Mahdian et al. (2007).

derive value from this impression type. As shown by Mehta et al. (2007), the competitiveness of the exponential averaging algorithm reaches $1 - \frac{1}{e}$ for $k \to \infty$ on this instance.

We evaluate the performance of our algorithm on this worst-case instance in Figure 8. Providing the optimum solution as prediction allows the algorithm to quickly ascend to a perfect robustness of 1. We also consider two (biased) corrupted versions of this prediction with $p \in \{50\%, 75\%\}$. In both cases, the algorithm still achieves a robustness above the competitiveness of the prediction. The dual base algorithm cannot deliver meaningful predictions as it only sees impressions of the first type, which are clearly not representative of the following impressions by construction.

## C.3 Evaluation of GAP on an Ad Words Instance

With an algorithm for GAP, we can also solve AdWords instances. This allows us to compare our generalized algorithm to the algorithm of Mahdian et al. (2007) under the same predictions. In Figure 9, we run both algorithms on synthetic instances from Section C.2 with an optimum prediction and random corruption ($p = 0.5$). Both algorithms seem to have similar consistency, but our algorithm achieves a better robustness, due to a different choice of constants in the underlying algorithms.

