# OpenReview forum: "Online Ad Allocation with Predictions"
_NeurIPS.cc/2023/Conference — NeurIPS 2023 poster_

### Official Review · Reviewer_eG65 · 2023-06-30

**Soundness:** 3 good
**Presentation:** 3 good
**Contribution:** 2 fair
**Rating:** 7
**Confidence:** 3

**Summary:**

This paper is for the online matching problem in the display advertising domain. Essentially there are shopper/supply side requesting coming in in a sequential order, and static advertiser side demand with budget constraint. The goal is to do some sort of global optimization (as opposed to greedy strategy to maximize value for each request t). The paper proposed a learning-augmented algorithm and demonstrated its effectiveness via a theorem as well as simulations.

Score changed from 6 to 7 after seeing the rebuttal from the authors.

**Strengths:**

1. The robustness against low-quality prediction. This is a very nice property to have for an algorithm to be practical. It seems the benefit doesn't vanish thought it decays. Given that the prediction quality is always something with uncertainty and affects the cumulative value for any algorithm, what is shown in this paper is already good.

2. Convincing simulation/experiment results using real-world dataset.

3. I particularly like line 170 - 191 which has done a nice job explaining the intuition of algorithm 1.

**Weaknesses:**

1. Don't know if the #impression over #advertiser ratio will significantly affect the simulation results, but I think this ratio are similar in those tested datasets. It's worth a little more discussion. I think for real-world scenario, the actual ratio won't be in the scale of 100.

2. I think the gain beyond the greedy algorithm that maximizes the single-impression utility is usually more important than the regret compared to OPT. Worth having related info in the paper.

3. How to handle the targeting constraint? This is very common in display advertising that an advertiser is only interested in a subset of impressions relevant to their own business. It effectively means one additional constraint in the primal problem.

4. The level of contribution beyond what is done in existing work. I didn't get the idea that Bama 2020 work forms this as an open problem though indeed they are relevant.

**Questions:**

see pros/cons

---

> ### Author Rebuttal · Authors · 2023-08-09
>
> We thank the reviewer for their insightful comments and now address the mentioned weaknesses.
>
> 1. This is an interesting point and we have now conducted further experiments with synthetic data using #impression to #advertiser ratios from 20 to 2000. We did obtain similar results for these ratios when scaling budgets for the advertisers proportionally (if the budget is too low or too high, the instance becomes very easy and the baseline exponential averaging algorithm manages to extract close to 100% of the optimum value). We will add such experiments and a discussion to a revised version of the paper. The plots can be found in the PDF that was attached to the global comment above in Figure 2.
>
> 2. We thank the reviewer for this suggestion. Following the reviewer's suggestion, we added greedy algorithms as baselines to our experiments. We ran experiments with two greedy schemes: One which considers only the impression values (Greedy), the other the gain after disposal (Discounted Greedy). The evaluation shows that the two greedy schemes do worse than the algorithm of Feldman et al. (2009a) in all instances, except for the Discounted Greedy algorithm on synthetic data. We provide the experimental results in the PDF that was attached to the global comment above in Figures 1 and 2. From a theoretical perspective, the competitive ratio with respect to the optimal solution provides the strongest guarantee and this coincides with the robustness in our setting.
>
> 3. Targeting constraints can be directly modeled by setting the value $w_{at}$ of impression $t$ and an advertiser $a$ without the need for additional constraints. For instance, if an impression $t$ is not relevant for advertiser $a$, we can set $w_{at}$ to any negative value, which will ensure that the impression $t$ is never allocated to advertiser $a$.
>
> 4. Our algorithm and analysis are a significant departure from prior works. More specifically:
>
>     - Compared to Metha et al. (2007), which is the most closely related algorithm with predictions, we tackle problems that have less structure, thus require the use of free disposal and foremost new algorithmic ideas to incorporate predictions. A detailed comparison can be found in Lines 200 through 218.
>
>     - Compared to Feldman et al. (2009a), which is the most closely related algorithm without predictions, we develop a novel analysis to prove consistency. Feldman et al. (2009a) analyzes their algorithm in a local manner by relating the changes in the primal and dual objective values after each individual update to the solutions, as it is common for algorithms following the primal-dual framework (see e.g. the survey of Buchbinder and Naor (2009)). This local approach is not sufficient to obtain our guarantee. Instead, we analyze our algorithm via a novel global argument that expresses the objective values of the prediction and the final solution constructed by the algorithm as suitable linear combinations of the impression values. A direct comparison of the coefficients in the resulting linear combinations will not give our guarantee. Instead, we make several important observations that allow us to redistribute mass across coefficients (see Lemmas 4 and 9 in the appendix). The resulting analysis is delicate, as we can see for example in the proof of Lemma 9 in the appendix. We believe our novel approach could potentially be used to obtain improved learning-augmented algorithms for other problems in the primal-dual framework, such as the ones detailed in the work of Buchbinder and Naor (2009).
>
>     - Bamas et al. (2020) develop learning-augmented algorithms for covering problems using the primal-dual method but leave packing problems for future work in their discussion on future directions. One of the main problems considered by Bamas et al. (2020) is set cover, which is a fundamentally different problem to ad allocation, both with and without predictions. Our works and theirs do not overlap beyond the general use of the primal-dual framework.

---

> > ### Comment · Reviewer_eG65 · 2023-08-16
> >
> > Thank you for your response and for addressing my questions especially on point 1 and 2, I will change my score to accept (from 6 to 7).

---

### Official Review · Reviewer_9H3m · 2023-07-06

**Soundness:** 3 good
**Presentation:** 3 good
**Contribution:** 2 fair
**Rating:** 6
**Confidence:** 3

**Summary:**

This paper studied Display Ads and the generalised assignment problem (GAP) where ad impressions arrive online and are allocated immediately to advertisers on the offline side. The difference between the two problems is that in Display Ads each impression takes uniform size of advertisers’ budgets, while in GAP the size is non-uniform. The paper considered the setting that incorporates predictions for allocation, which can be accessed by the algorithm upon arrival of each impression. The paper proposed a primal-dual algorithm based on Feldman et al. (2009a), with new ingredients to deal with predictions. Firstly, the new algorithm decides whether to allocate an impression to the predicted advertiser or to one with max discounted gain based on comparison involving a robustness-consistency trade-off parameter. Secondly, the algorithm updates the dual variable with an exponential mechanism involving the trade-off parameter. The paper provided theoretical analysis for the robustness and consistency factors, as well as experimental results on synthetic and real world datasets.

**Strengths:**

1. Online matching and related problems have been extensively studied in the literature and found their applications in online business. This paper studied Display Ads and GAP, which are generalized version of online bipartite matching and Ad Words, that could capture more complicated scenarios in practice. Worst-case guarantees can be restrictive, while external or historical information may be available for making online decisions. This paper proposed an efficient and effective algorithm in this setting.

2. The paper proposed a solid primal-dual approach with theoretical performance guarantee that incorporates predictions that is accessible in general forms. The performance of the algorithm improves over worst-case and random mixture baselines in experiments.

3. The paper is clearly written. The main text provides necessary intuition and discussion about the algorithm and analysis, as well as comparison with related work.

**Weaknesses:**

1. The paper only presented the proposed approach for Display Ads in the main text, but put the whole GAP part to Appendix. It would be helpful to briefly discuss in the main text how to extend the approach to GAP and take care of general impression sizes.

2. It is not clear if consistency is a good metric for evaluating the performance of algorithms with predictions. Unlike robustness, the consistency factor relies on a varying value of solution PRD to be compared against. For instance, in the upper left of Figure 3, the consistency value is high when PRD itself has low competitive ratio, while it is below 1 when PRD=OPT. It is hard to relate the absolute value of consistency to the quality of ALG.

3. The paper mentioned that the algorithm incorporates machine-learning predictions in abstract and introduction, which seems misleading. The actual algorithm receives a prediction PRD(t) for each impression t, which is presented in an abstract form and is not necessarily related to machine learning.

**Questions:**

1. In Theorem 1 the robustness R is a decreasing function with respect to \alpha, but in experiments the robustness has a trend of (slightly) increasing w.r.t. \alpha. What is the cause of this phenomenon?

2. The limit of R(\alpha) in Theorem 1 tends to be 1 when \alpha tends to be 0. How does R(\alpha) compare with the worst-case ratio 1-1/e?

**Limitations:**

The paper briefly discussed limitations of the proposed algorithm.

---

> ### Author Rebuttal · Authors · 2023-08-09
>
> We thank the reviewer for their helpful comments and address the weaknesses in the following.
>
> 1. We thank the reviewer for this suggestion and we will add such a discussion to the main body. Our reason for discussing only Display Ads in the main body is that it contains some of the most important algorithmic ideas of our approach.
>
> 2. Our work follows the standard practice in the literature on learning-augmented algorithms to measure the performance of an algorithm via consistency and robustness. We agree with the reviewer that when the prediction is good, the consistency allows us to gauge how closely the algorithm follows it. For worse predictions, it is most informative to look at both consistency and robustness, as this tells us how well the algorithm is able to overcome errors in the prediction.
>
> 3. It is true that predictions can come from any source and we will clarify this in a revised version of our paper.
>
> We now address the reviewer's questions.
>
> 1. This is an interesting point an we will add an explanation for this behavior to a revised version of our paper. In our theoretical analysis, we have to assume an adversarial input from our prediction, which means that the robustness necessarily declines the more we trust the prediction. In practice, a prediction is not chosen to be adversarial, even though its objective value can be much less than the optimum. In our experiments, we observe that for increasing $\alpha$, our algorithm is able to exploit “good suggestions” more while it still ignores “bad suggestions”. This explains why there is no immediate trade-off as in the guarantees of Theorem 1 and showcases the practical merit of combining worst-case algorithms with learned predictions, which may still err on some parts of the input.
>
> 2. In our paper, we only consider parameters $\alpha\ge1$. For $\alpha=1$ (which means that we do not trust the prediction at all), our algorithm is identical to the worst-case baseline due to Feldman et al. (2009a). In this case, we also recover the worst-case competitive ratio of $1-\frac{1}{e}$ in the large-budget case.

---

### Official Review · Reviewer_2o36 · 2023-07-10

**Soundness:** 2 fair
**Presentation:** 2 fair
**Contribution:** 2 fair
**Rating:** 4
**Confidence:** 3

**Summary:**

The paper discusses the problem of ad allocation and its generalization, the generalized assignment problem (GAP), which are two well-studied online packing problems with important applications in ad allocation and other areas. The paper presents an algorithm for both problems that incorporate machine-learned predictions and can thus improve the performance beyond the worst-case. The algorithm is based on the work of Feldman et al. (2009a) and similar in nature to Mahdian et al. (2007) who were the first to develop a learning-augmented algorithm for the related, but more structured Ad Words problem. The paper’s contributions are that it designs the first algorithms that incorporate machine-learned predictions for Display Ads and GAP. The two problems are general online packing problems that capture a wide range of applications. The algorithm follows a primal-dual approach, which yields a combinatorial algorithm that is very efficient and easy to implement. It is able to leverage predictions which can be learned from historical data. Using a novel analysis, the paper shows that the algorithm is robust against bad predictions and able to improve its performance with good predictions. In particular, it is able to bypass the strong lower bound on the worst-case competitive ratio for these problems. The paper experimentally verifies the practical applicability of its algorithm under various kinds of predictions on synthetic and real-world data sets. Here, it observes that its algorithm is able to outperform the baseline worst-case algorithm and the random-mixture algorithm.

**Strengths:**

* Originality: lies in its use of machine-learned predictions to improve the performance of online packing algorithms for ad allocation, i.e., Display Ads and GAP.
* Quality: presents a rigorous analysis of its algorithm and provides experimental results that demonstrate its effectiveness.
* Significance: addresses an important problem in online advertising and provides a novel solution that can improve the performance of existing algorithms.

**Weaknesses:**

1. The paper does not provide a detailed analysis of the limitations of its algorithm. For example, it is not clear how the algorithm would perform in situations where the predictions are inaccurate or where there are other sources of uncertainty.
2. The paper does not provide a comparison of its algorithm with other state-of-the-art algorithms for online ad allocation. This makes it difficult to assess the relative performance of the proposed algorithm and to determine whether it is truly novel and effective.
3. The paper does not provide a detailed discussion of the practical implications of its algorithm for real-world ad allocation systems. This could be addressed in future work by providing a more detailed analysis of the computational requirements and scalability of the proposed algorithm, as well as by conducting experiments on real-world data sets.

**Questions:**

See the Weakness section, especially Weakness 2.

**Limitations:**

See the Weakness section, especially Weakness 1.

---

> ### Author Rebuttal · Authors · 2023-08-09
>
> We thank the reviewer for their comments and address the mentioned weaknesses in the following.
>
> 1. Our algorithm is specifically designed to perform well even if the predictions are inaccurate (from any source of uncertainty). In the field of learning-augmented algorithms, the measure of robustness is used to indicate how well the algorithm performs under an arbitrarily bad prediction. We show a theoretical bound on the robustness in Theorem 1 and experimentally validate our algorithm on inaccurate predictions in Section 4.
>
> 2. To the best of our knowledge, we compare our algorithm against the state-of-the art algorithms with theoretical guarantees for the problems we consider, both with and without predictions. We are the first to develop learning-augmented algorithms for the problems Display Ads and GAP. For these problems, we compare against the state-of-the-art worst-case algorithm, which is due to Feldman et al. (2009a). This work achieves a competitive ratio of $1-\frac{1}{e}$ which is best possible in the worst-case. For the more structured Ad Words problem, to the best of our knowledge, the best known competitive ratio with predictions is achieved by the algorithm of Mehta et al. (2007). We evaluate our algorithm on Ad Words instances and compare with this work in Section C.3 of the appendix in the supplementary materials. We will reference this in the main body. We would appreciate any suggestions from the reviewer on other algorithms and we will try to add them to a revised version of our paper.
>
> 3. We do provide experiments on the real-world datasets Yahoo and iPinYou in Section 4 (Line 293) of our paper. These are the two main datasets that are publicly available. If the reviewer has suggestions for other datasets that are suitable, we are happy to try to evaluate our algorithms on these datasets. Our algorithm is very efficient and scalable, and we ran all our experiments on a laptop with an i7-1165G7 CPU and 16Gb of memory. We are happy to add a discussion on the scalability of the algorithm in the next revision. We agree with the reviewer that exploring the implications of our work on real-world ad allocation systems is an interesting direction for future work.

---

### Official Review · Reviewer_3Ccp · 2023-07-10

**Soundness:** 3 good
**Presentation:** 3 good
**Contribution:** 3 good
**Rating:** 7
**Confidence:** 4

**Summary:**

This paper considers the online display ads and generalized assignment problems under free disposal in the "algorithms with predictions setting".  In the display ads problem there are offline advertisers $a$ with budgets $B_a$.  A sequence of impressions arrive online with differing values to each advertiser (impression $t$ has value $w_{at}$ to advertiser $a$).  The algorithm must either allocate each impression to an advertiser, earning its value and consuming one unit of the advertiser's capacity, or reject the impression entirely.  Decisions are made irrevocably, but in the free disposal model the algorithm may assign more impressions to an advertiser than their capacity allows but the algorithm only gains the value of the most valuable impressions that fit within the capacity.  The generalized assignment problem generalizes the display ads problem by allowing impression $t$ to instead consume $u_{at}$ units of advertiser $a$'s capacity.

In the "algorithms with predictions" setting the online algorithm also has access to potentially noisy predictions about the optimal solution.  In particular, this paper assumes that the algorithm may access a prediction $PRD(t)$ when impression $t$ arrives which gives a hint about which advertiser to allocate impression $t$ to in an optimal solution in hindsight.  The prediction may be incorrect, so two quantities are analyzed: consistency and robustness.  Consistency is given by the worst-case ratio of $ALG/PRD$, where $ALG$ is the algorithms solution quality and $PRD$ is the predictions solution quality.  Robustness is given by the worst -case ratio of $ALG/OPT$, where $OPT$ is the optimal solution in hindsight.

Theoretically, this paper gives algorithms with non-trivial tradeoffs between robustness and consistency.  The algorithm is based off of the primal-dual algorithm used by Feldman et al. 2009, but modified to account for the predictions.  Proving their consistency bound needed new techniques since a local analysis as is typically used in the primal-dual method is insufficient here.

The authors complement the theoretical results with an experimental analysis on both real and synthetic data of their proposed algorithm using various predictions as input.  The experiments compare to the worst-case baseline of Feldman et al. 2009 and the simple random-mixture algorithm they discuss in the introduction.





**Strengths:**

This paper considers online allocation problems motivated by advertising applications in the algorithms with predictions setting.  Prior work has considered related problems in this setting.  This paper considers more general versions of these problems and also considers the free-disposal setting, which has not yet been considered in the algorithms with predictions literature.  The paper gives strong theoretical guarantees for their algorithm and complements this with a thorough experimental analysis.

**Weaknesses:**

 - Some of the presentation of the results in the experiment section could be clarified/improved.  See comments/questions below.
 - The improvement over the worst-case baseline using "more realistic" predictions is somewhat narrow (although in practice this smaller improvement could matter in some cases).
 - Theoretically, it is not clear if the trade-off between robustness and consistency is tight.  Giving tight guarantees has become of interest recently, see the two references below:
    - Wei, A., Zhang, F., "Optimal Robustness-Consistency Trade-offs for Learning-Augmented Online Algorithms."  NeurIPS 2020.
    - Jin, B., Ma, W., "Online Bipartite Matching with Advice: Tight Robustness-Consistency Tradeoffs for the Two-Stage Model."  NeurIPS 2022.  (this paper may also be good to discuss in the related work).




**Questions:**

## Questions

 - Can you

 - Please clarify what is meant by "prediction competitiveness" in the plots in Figure 4.

 - What was observed experimentally for the competitive ratio of the worst case baseline?

 - In Figures 3 and 5 the average competitive ratio across 5 trials is reported in parentheses for different predictors.  Since there is a hypereparameter $\alpha$ that is varied can you clarify which value of $\alpha$ was used to produce these numbers?

## Comments

 - For the plots in figure 2, it might be helpful to visually compare with the consistency/robustness guarantees that would be guaranteed by the random-mixture algorithm as well as the algorithm due to Mahdian et al. 2007.


**Limitations:**

I believe the authors have adequately addressed potential limitations and there is not a high potential for negative societal impacts from this work.

---

> ### Author Rebuttal · Authors · 2023-08-09
>
> We thank the reviewer for their insightful comments and address the weaknesses in the following.
>
> 1. We thank the reviewer for the suggestions for improvements. We will incorporate them in the paper.
>
> 2. Indeed, the improvement over the worst-case algorithm when using the Dual Base prediction is small. We discuss the performance of our algorithm with the Dual Base prediction in Section C.2 under “Results”. We observed in all of our datasets, both in real-world and synthetic instances, that the Dual Base prediction is a poorly performing algorithm despite its strong theoretical guarantees (e.g. on the iPinyYou dataset, Dual Base achieves a competitive ratio of 64% while the worst-case algorithm of Feldman et al (2009a) achieves 95%). We speculate that the reason is that it constructs a fixed allocation rule based only on an initial sample. Moreover, we observe that the errors of the Dual Base algorithm may mislead our algorithm. We observe that the better performing predictors which we obtain by corruptions of the optimum lead to much better results.
>
> 3. We thank the reviewer for the suggestion and the references. We will cover the references and include a discussion on tightness in the revision. Our analysis technique can be brought further to achieve a slight improvement in the guarantees. However, doing so is quite technical and we omitted it in the interest of simplicity and conciseness. We think it is a interesting direction for future work to see whether a different analysis can yield more significant improvements.
>
> We also address the reviewer's questions.
>
> 1. The prediction competitiveness is the ratio of the objective value of the predicted solution and the optimum, i.e. $\mathrm{PRD}/\mathrm{OPT}$. We will add this to the description.
>
> 2. The competitive ratio of the worst-case baseline is 95.8\% on the iPinYou instance, 87.6\% on the Yahoo instance in Figure 3, and 90.6\% for the synthetic instance in Figure 5. The black line in the plots shows the robustness of the worst-case baseline, which coincides with the competitive ratio.
>
> 3. The competitive ratio of the predictors are independent of $\alpha$. The value of $\alpha$ is used only by our algorithm, alongside a predicted solution.
>
> We thank the reviewer for the comment on Figure 2 and we will add the other guarantees to a revised version of our paper.

---

> > ### Comment · Reviewer_3Ccp · 2023-08-16
> >
> > Thank you for your response, I will keep my score the same (accept).

---

### Author Rebuttal · Authors · 2023-08-09

Following reviewer eG65's suggestions, we ran additional experiments with two additional baselines and varied the number of impressions in our synthetic instances. The two additional baselines are two greedy schemes: One considers only the impression values (Greedy) and the other the gain after disposal (Discounted Greedy). The results can be found in the attached PDF.

---

### Decision · Program_Chairs · 2023-09-21

**Decision:**

Accept (poster)

**Comment:**

This paper gives strong theoretical guarantees for using predictions in online allocation problems. The reviewers felt this work went beyond prior work leveraging new algorithmic predictions and extending prior models. Overall, this was viewed as a novel contribution.